**A General Analytical Model for Head Response to Oscillatory Pumping in**
**Unconfined Aquifers: Effects of Delayed Gravity Drainage and Initial**
**Condition**
**Ching-Sheng Huang[a], Ya-Hsin Tsai[b], Hund-Der Yeh[b*] and Tao Yang[a*]**
[a] State Key Laboratory of Hydrology-Water Resources and Hydraulic Engineering, Center for
Global Change and Water Cycle, Hohai University, Nanjing 210098, China
[b] Institute of Environmental Engineering, National Chiao Tung University, Hsinchu 300,
Taiwan
[*] Corresponding authors:
Hund-Der Yeh; E-mail: hdyeh@mail.nctu.edu.tw; Tel.: +886-3-5731910; fax: +886-3-

11   5725958

Tao Yang; E-mail: tao.yang@hhu.edu.cn; Tel.: +86-13770918075
Submission to *Hydrology and Earth System Sciences* on 11 September 2018
Re-submission to *Hydrology and Earth System Sciences* on 24 December 2018
Re-re-submission to *Hydrology and Earth System Sciences* on 27 February 2019
**Key points**
1.   An analytical model of the hydraulic head due to oscillatory pumping in unconfined

18        aquifers is presented.

2.   Head fluctuations affected by instantaneous and delayed gravity drainages are discussed.
3.   The effect of initial condition on the phase of head fluctuation is analyzed.
4.   The present solution agrees well to head fluctuation data taken from a field oscillatory

22        pumping.

# **Abstract**

Oscillatory pumping tests (OPTs) provide an alternative to constant-head and constant-rate
pumping tests for determining aquifer hydraulic parameters when OPT data are analyzed based
on an associated analytical model coupled with an optimization approach. There is a large
number of analytical models presented for the analysis of OPT. The combined effects of
delayed gravity drainage (DGD) and initial condition regarding the hydraulic head are
commonly neglected in the existing models. This study aims to develop a new model for
describing the hydraulic head fluctuation induced by OPT in an unconfined aquifer. The model
contains a groundwater flow equation with the initial condition of static water table, Neumann
boundary condition specified at the rim of a partially screened well, and a free surface equation
describing water table motion with the DGD effect. The solution is derived using the Laplace,
finite-integral, and Weber transforms. Sensitivity analysis is carried out for exploring head
response to the change in each of hydraulic parameters. Results suggest the DGD reduces to
instantaneous gravity drainage in predicting transient head fluctuation when dimensionless
parameter $a_1 = \epsilon S_y b / K_z$ exceeds 500 with empirical constant $\epsilon$, specific yield $S_y$, aquifer
thickness $b$, and vertical hydraulic conductivity $K_z$. The water table can be regarded as a no-
flow boundary when $a_1 < 10^{-2}$ and $P < 10^4$ s with $P$ being the period of oscillatory
pumping rate. A pseudo-steady state model without initial condition causes a time shift from
the actual transient model in predicting simple harmonic motion of head fluctuation during a
late pumping period. In addition, the present solution agrees well to head fluctuation data
observed at the Savannah River site.
**KEYWORDS:** oscillatory pumping test, analytical solution, free surface equation, delayed
gravity drainage, initial condition

# Notation and Abbreviation

| | |
|---|---|
| $a$ | $bD_r/(C_y r_w^2)$ |
| $a_1, a_2$ | $b/(\kappa C_y)$, $r_w^2/(\kappa D_r)$ |
| $b$ | Aquifer thickness |
| $C_y$ | $K_z/S_y$ |
| $D_r$ | $K_r/S_s$ |
| DGD | Delayed gravity drainage |
| $h$ | Hydraulic head |
| $\bar{h}$ | Dimensionless Hydraulic head, i.e., $\bar{h} = 2\pi l K_r h/Q$ |
| IGD | Instantaneous gravity drainage |
| $K_r, K_z$ | Aquifer horizontal and vertical hydraulic conductivities, respectively |
| LHS | Left-hand side |
| $l$ | Screen length, i.e., $z_u - z_l$ |
| OPT | oscillatory pumping test |
| $P$ | Period of oscillatory pumping rate |
| PSS | Pseudo-steady state |
| $\bar{P}$ | Dimensionless period, i.e., $\bar{P} = D_r P/r_w^2$ |
| $p$ | Laplace parameter |
| $Q$ | Amplitude of oscillatory pumping rate |
| RHS | Right-hand side |
| $r$ | Radial distance from the center of pumping well |
| $\bar{r}$ | Dimensionless radial distance, i.e., $\bar{r} = r/r_w$ |
| $r_w$ | Radius of pumping well |
| SHM | Simple harmonic motion |
| $S_s, S_y$ | Specific storage and specific yield, respectively |
| $t$ | Time since pumping |
| $\bar{t}$ | Dimensionless pumping time, i.e., $\bar{t} = D_r t/r_w^2$ |
| $z$ | Elevation from aquifer bottom |
| $z_l, z_u$ | Lower and upper elevations of well screen, respectively |
| $\bar{z}$ | Dimensionless elevation, i.e., $\bar{z} = z/b$ |
| $\bar{z}_l, \bar{z}_u$ | $z_l/b, z_u/b$ |
| $\alpha$ | $K_z/K_r$ |
| $\beta_n$ | Roots of Eq. (15) |
| $\kappa$ | $1/\epsilon$ |
| $\gamma$ | Dimensionless frequency of oscillatory pumping rate, i.e., $\omega r_w^2/D_r$ |
| $\epsilon$ | Empirical constant associated with delayed gravity drainage |
| $\mu$ | $\alpha r_w^2/b^2$ |
| $\omega$ | Frequency of oscillatory pumping rate, i.e., $\omega = 2\pi/P$ |

## 1. Introduction

Numerous attempts have been made by researchers to the study of oscillatory pumping test (OPT) that is an alternative to constant-rate and constant-head pumping tests for determining aquifer hydraulic parameters (e.g., Vine et al., 2016; Christensen et al., 2017; Watlet et al., 2018). The concept of OPT was first proposed by Kuo (1972) in the petroleum literature. The process of OPT contains extraction stages and injection stages. The pumping rate, in other words, varies periodically as a sinusoidal function of time. Compared with traditional constant-rate pumping, OPT in contaminated aquifers has the following advantages: (1) low cost because of no disposing contaminated water from the well, (2) reduced risk of treating contaminated fluid, (3) smaller contaminant movement, and (4) stable signal easily distinguished from background disturbance such as tide effect and varying river stage (e.g., Spane and Mackley, 2011). However, the disadvantages of OPT include the need of an advanced apparatus producing periodic rate. Oscillatory hydraulic tomography adopts several oscillatory pumping wells with different frequencies (e.g., Yeh and Liu, 2000; Cardiff et al., 2013; Zhou et al., 2016; Muthuwatta et al., 2017). Aquifer heterogeneity can be mapped by analyzing multiple data collected from observation wells. Cardiff and Barrash (2011) reviewed articles associated with hydraulic tomography and classified them according to nine categories in a table.

Various groups of researchers have worked with analytical and numerical models for OPT; each group has its own model and investigation. For example, Black and Kipp (1981) assumed the response of confined flow to OPT as simple harmonic motion (SHM) in the absence of initial condition. Cardiff and Barrash (2014) built an optimization formulation strategy using the Black and Kipp analytical solution. Dagan and Rabinovich (2014) also assumed hydraulic head fluctuation as SHM for OPT at a partially screened well in unconfined aquifers. Cardiff et al. (2013) characterized aquifer heterogeneity using the finite element-based COMSOL software that adopts SHM hydraulic head variation for OPT. On the other hand, Rasmussen et al. (2003) found hydraulic head response tends to SHM at a late period of pumping time when

considering initial condition prior to OPT. Bakhos et al. (2014) used the Rasmussen et al. (2003)
analytical solution to quantify the time after which hydraulic head fluctuation can be regarded
as SHM since OPT began. As mentioned above, most of the models for OPT assume hydraulic
head fluctuation as SHM without initial condition, and all of them treat the pumping well as a
line source with infinitesimal radius.
Field applications of OPT for determining aquifer parameters have been conducted in
recent years. Rasmussen et al. (2003) estimated aquifer hydraulic parameters based on 1- or 2-
hour period of OPT at the Savannah River site. Maineult et al. (2008) observed spontaneous
potential temporal variation in aquifer diffusivity at a study site in Bochum, Germany. Fokker
et al. (2012; 2013) presented spatial distributions of aquifer transmission and storage
coefficient derived from curve fitting based on a numerical model and field data from
experiments at the southern city-limits of Bochum, Germany. Rabinovich et al. (2015)
estimated aquifer parameters of equivalent hydraulic conductivity, specific storage and specific
yield at the Boise Hydrogeophysical Research Site by curve fitting based on observation data
and the Dagan and Rabinovich (2014) analytical solution. They conclude the equivalent
hydraulic parameters can represent the actual aquifer heterogeneity of the study site.
Although a large number of studies have been made in developing analytical models for
OPT, little is known about the combined effects of delayed gravity drainage (DGD), finite-
radius pumping well, and initial condition prior to OPT. Analytical solution to such a question
will not only have important physical implications but also shed light on OPT model
development. This study builds an improved model describing hydraulic head fluctuation
induced by OPT in an unconfined aquifer. The model is composed of a typical flow equation
with the initial condition of static water table, an inner boundary condition specified at the rim
of the partially screened well for incorporating finite-radius effect, and a free surface equation
describing the motion of water table with the DGD effect. The analytical solution of the model
is derived by the methods of Laplace transform, finite-integral transform, and Weber transform.
Based on the present solution, sensitivity analysis is performed to explore the hydraulic head
in response to the change in each of hydraulic parameters. The effects of DGD and
instantaneous gravity drainage (IGD) on the head fluctuations are compared. The quantitative
criterion for treating the well radius as infinitesimal is discussed. The effect of the initial
condition on the phase of head fluctuation is investigated. In addition, curve fitting of the
present solution to head fluctuation data recorded at the Savannah River site is presented.
**2. Methodology**
**2.1. Mathematical model**
Consider an OPT in an unconfined aquifer illustrated in Fig. 1. The aquifer is of unbound lateral
extent with a finite thickness $b$. The radial distance from the centerline of the well is $r$; an
elevation from the impermeable bottom of the aquifer is $z$. The well with outer radius $r_w$ is
screened from elevation $z_u$ to $z_l$.

The flow equation describing spatiotemporal head distribution in aquifers can be written

as:
$D_r \left( \frac{\partial^2 h}{\partial r^2} + \frac{1}{r} \frac{\partial h}{\partial r} + \alpha \frac{\partial^2 h}{\partial z^2} \right) = \frac{\partial h}{\partial t}$   for   $r_w \leq r < \infty, \ 0 \leq z \leq b$ and $t \geq 0$   (1)
where $D_r = K_r / S_s$; $\alpha = K_z / K_r$; $h(r, z, t)$ is hydraulic head at location ($r$; $z$) and time $t$; $K_r$
and $K_z$ are respectively the radial and vertical hydraulic conductivities; $S_s$ is the specific
storage. Consider water table as a reference datum where the elevation head is set to zero; the
initial condition is expressed as:
$h = 0$ at $t = 0$   (1)
The rim of the wellbore is regarded as an inner boundary under the Neumann condition
expressed as:
$2\pi r_w K_r l \frac{\partial h}{\partial r} = \begin{cases} Q\sin(\omega t) \text{ for } z_l \leq z \leq z_u \\ 0 \text{ outside screen interval} \end{cases}$ at $r = r_w$   (2)
where $l = z_u - z_l$ is screen length; $Q$ and $\omega = 2\pi/P$ are respectively the amplitude and
frequency of oscillatory pumping rate (i.e., $Q\sin(\omega t)$) with a period $P$. Water table motion can
be defined by Eq. (4a) for IGD (Neuman, 1972) and Eq. (4b) for DGD (Moench, 1995).
$\frac{\partial h}{\partial z} = -\frac{1}{C_y}\frac{\partial h}{\partial t}$   at   $z = b$   for IGD        (3a)
$\frac{\partial h}{\partial z} = \frac{1}{\kappa C_y}\int_0^t \frac{\partial h}{\partial \tau}\exp(-(t-\tau)/\kappa)\,d\tau$   at   $z = b$   for DGD        (4b)
where $C_y = K_z/S_y$, $\kappa = 1/\epsilon$ with $\epsilon$ being an empirical constant, and $S_y$ is the specific
yield. Note that Eq. (4b) reduces to Eq. (4a) when $\kappa \to \infty$ or $\epsilon = 0$. The impervious aquifer
bottom is under the no-flow condition:
$\frac{\partial h}{\partial z} = 0$ at $z = 0$        (4)
The hydraulic head far away from the pumping well remains constant, written as
$\lim_{r\to\infty} h(r,z,t) = 0$        (5)

Define dimensionless variables and parameters as follows:

$\bar{h} = \frac{2\pi l K_r}{Q}h$, $\bar{r} = \frac{r}{r_w}$, $\bar{z} = \frac{z}{b}$, $\bar{z}_l = \frac{z_l}{b}$, $\bar{z}_u = \frac{z_u}{b}$, $\bar{t} = \frac{D_r}{r_w^2}t$, $\bar{\tau} = \frac{D_r}{r_w^2}\tau$, $\bar{P} = \frac{D_r}{r_w^2}P$
$\gamma = \frac{\omega r_w^2}{D_r}$, $\mu = \frac{\alpha r_w^2}{b^2}$, $a = \frac{bD_r}{C_y r_w^2}$, $a_1 = \frac{b}{\kappa C_y}$, $a_2 = \frac{r_w^2}{\kappa D_r}$        (6)
where the overbar stands for a dimensionless symbol. Note that the magnitude of $a_1$ is related
to the DGD effect (Moench, 1995) and $\gamma$ is a dimensionless frequency parameter. With Eq. (7),
the dimensionless forms of Eqs. (1) - (6) become, respectively,
$\frac{\partial^2 \bar{h}}{\partial \bar{r}^2} + \frac{1}{\bar{r}}\frac{\partial \bar{h}}{\partial \bar{r}} + \mu\frac{\partial^2 \bar{h}}{\partial \bar{z}^2} = \frac{\partial \bar{h}}{\partial \bar{t}}$ for $1 \leq \bar{r} < \infty$, $0 \leq \bar{z} < 1$ and $\bar{t} \geq 0$        (7)
$\bar{h} = 0$ at $\bar{t} = 0$        (8)
$\frac{\partial \bar{h}}{\partial \bar{r}} = \begin{cases} \sin(\gamma\bar{t}) & \text{for } \bar{z}_l \leq \bar{z} \leq \bar{z}_u \\ 0 & \text{outside screen interval} \end{cases}$ at $\bar{r} = 1$        (9)
$\frac{\partial \bar{h}}{\partial \bar{z}} = -a\frac{\partial \bar{h}}{\partial \bar{t}}$ at $\bar{z} = 1$ for IGD        (10a)
$\frac{\partial \bar{h}}{\partial \bar{z}} = -a_1\int_0^{\bar{t}} \frac{\partial \bar{h}}{\partial \bar{\tau}}\exp(-a_2(\bar{t}-\bar{\tau}))\,d\bar{\tau}$ at $\bar{z} = 1$ for DGD        (11b)
$\frac{\partial \bar{h}}{\partial \bar{z}} = 0$ at $\bar{z} = 0$        (12)
$\lim_{\bar{r}\to\infty} \bar{h}(\bar{r},\bar{z},\bar{t}) = 0$        (13)
Eqs. (8) – (13) represent the transient DGD model when excluding (11a) and transient IGD
model when excluding (11b).

## 2.2. Transient solution for unconfined aquifer

The Laplace transform and finite-integral transform are applied to solve Eqs. (8) - (13)
(Latinopoulos, 1985; Liang et al., 2017; 2018). The transient solution can then be expressed as
$\bar{h}(\bar{r}, \bar{z}, \bar{t}) = \bar{h}_{\exp}(\bar{r}, \bar{z}, \bar{t}) + \bar{h}_{\mathrm{SHM}}(\bar{r}, \bar{z}, \bar{t})$ (14a)
with
$\bar{h}_{\exp}(\bar{r}, \bar{z}, \bar{t}) = \frac{-2\gamma}{\pi} \sum_{n=1}^{\infty} \int_0^{\infty} \cos(\beta_n \bar{z}) \exp(p_0 \bar{t}) \, \mathrm{Im}(\varepsilon_1 \varepsilon_2) \, d\zeta$ (14b)
$\bar{h}_{\mathrm{SHM}}(\bar{r}, \bar{z}, \bar{t}) = \bar{A}_t(\bar{r}, \bar{z}) \cos(\gamma \bar{t} - \phi_t(\bar{r}, \bar{z}))$ (14c)
$\bar{A}_t(\bar{r}, \bar{z}) = \sqrt{a_t(\bar{r}, \bar{z})^2 + b_t(\bar{r}, \bar{z})^2}$ (14d)
$a_t(\bar{r}, \bar{z}) = \frac{2}{\pi} \sum_{n=1}^{\infty} \int_0^{\infty} p_0 \cos(\beta_n \bar{z}) \, \mathrm{Im}(\varepsilon_1 \varepsilon_2) \, d\zeta$ (14e)
$b_t(\bar{r}, \bar{z}) = \frac{2\gamma}{\pi} \sum_{n=1}^{\infty} \int_0^{\infty} \cos(\beta_n \bar{z}) \, \mathrm{Im}(\varepsilon_1 \varepsilon_2) \, d\zeta$ (14f)
$\phi_t(\bar{r}, \bar{z}) = \cos^{-1}(b_t(\bar{r}, \bar{z})/\bar{A}_t(r, \bar{z}))$ (14g)
$\varepsilon_1 = K_0(\lambda_0 \bar{r})(\sin(\bar{z}_u \beta_n) - \sin(\bar{z}_l \beta_n))/(\beta_n \lambda_0 K_1(\lambda_0)(p_0^2 + \gamma^2))$ (14h)
$\varepsilon_2 = (\beta_n^2 + c_0^2)/(\beta_n^2 + c_0^2 + c_0)$ (14i)
$p_0 = -\zeta - \mu\beta_n^2$ (14j)
$\lambda_0 = \sqrt{\zeta} i$ (14k)
where $c_0 = a p_0$ for IGD and $a_1 p_0/(p_0 + a_2)$ for DGD, $i$ is the imaginary unit, Im(-) is the
imaginary part of a complex number, $K_0(-)$ and $K_1(-)$ are the modified Bessel functions
of the second kind of order zero and one, respectively, and $\beta_n$ is the positive roots of the
equation:
$\tan \beta_n = c_0/\beta_n$ (15)
The method to find the roots of $\beta_n$ is discussed in Section 2.3. The detailed derivation of
Eqs. (14a) – (14k) is presented in the supporting material. The first term on the right-hand side
(RHS) of Eq. (14a) exhibits exponential decay due to the initial condition since pumping began;
the second term defines SHM with amplitude $\bar{A}_t(\bar{r}, \bar{z})$ and phase shift $\phi_t(\bar{r}, \bar{z})$ at a given
point $(\bar{r}, \bar{z})$. The numerical results of the integrals in Eqs. (14b), (14e) and (14f) are obtained
by the Mathematica NIntegrate function.

## 175    2.3. Calculation of $\beta_n$

The eigenvalues $\beta_1$, …, $\beta_n$, the roots of Eq. (15) can be determined by applying the
Mathematica function FindRoot based on Newton's method with reasonable initial guesses.
The roots are located at the intersection of the curves plotted by the RHS and left-hand side
(LHS) functions of $\beta_n$ in Eq. (15). The roots are very close to the vertical asymptotes of the
periodical tangent function $\tan \beta_n$. When $c_0 = ap_0$, the initial guess for each $\beta_n$ can be
considered as $\beta_{0,n} + \delta$ where $\beta_{0,n} = (2n - 1)\pi/2$, $n \in (1,2, \dots \infty)$ and $\delta$ is a small
positive value set to $10^{-10}$. When $c_0 = a_1 p_0/(p_0 + a_2)$, the initial guess is set to $\beta_{0,n} - \delta$ for
$a_2 - \zeta \leq 0$. There is an additional vertical asymptote at $\beta_n = \sqrt{(a_2 - \zeta)/\mu}$ derived from the
RHS function of Eq. (15) (i.e., $p_0 + a_2 = 0$) if $a_2 - \zeta > 0$. The initial guess is therefore set
to $\beta_{0,n} + \delta$ for $\beta_{0,n}$ on the LHS of the asymptote and $\beta_{0,n} - \delta$ for $\beta_{0,n}$ on the RHS.

## 186    2.4. Transient solution for confined aquifer

When $S_y = 0$ (i.e., $a = 0$ or $a_1 = 0$), Eq. (11a) or (11b) reduces to $\partial \bar{h}/\partial \bar{z} = 0$ for no-flow
condition at the top of the aquifer, indicating the unconfined aquifer becomes a confined one.
Under this condition, Eq. (15) becomes $\tan \beta_n = 0$ with roots $\beta_n = 0$, $\pi$, $2\pi$, …, $n\pi$, …,
$\infty$; Eq. (14i) reduces to $\varepsilon_2 = 1$; factor 2 in Eqs. (14b), (14e) and (14f) is replaced by unity for
$\beta_n = 0$ and remains for the others. The analytical solution of the transient head for the
confined aquifer can be expressed as Eqs. (14a) - (14k) with
$$\bar{h}_{\exp}(\bar{r}, \bar{z}, \bar{t}) = \frac{-\gamma}{\pi} \int_0^\infty \text{Im}(\varepsilon_0) \exp(-\zeta \bar{t}) \, d\zeta - \frac{2\gamma}{\pi} \sum_{n=1}^\infty \int_0^\infty \cos(n\pi \bar{z}) \, \text{Im}(\varepsilon_1) \exp(p_0 \bar{t}) \, d\zeta$$

(16a)

$a_t(\bar{r}, \bar{z}) = -\frac{1}{\pi} \int_0^\infty \zeta \text{Im}(\varepsilon_0) \, d\zeta + \frac{2}{\pi} \sum_{n=1}^\infty \int_0^\infty p_0 \cos(n\pi \bar{z}) \, \text{Im}(\varepsilon_1) \, d\zeta$                  (16b)
$b_t(\bar{r}, \bar{z}) = \frac{\gamma}{\pi} \int_0^\infty \text{Im}(\varepsilon_0) \, d\zeta + \frac{2\gamma}{\pi} \sum_{n=1}^\infty \int_0^\infty \cos(n\pi\bar{z}) \, \text{Im}(\varepsilon_1) \, d\zeta$          (16c)
$\varepsilon_0 = (\bar{z}_u - \bar{z}_l) K_0(\lambda_0 \bar{r}) / (\lambda_0 K_1(\lambda_0)(\zeta^2 + \gamma^2))$          (16d)

Note that Eq. (14h) reduces to Eq. (16d) based on $\beta_n = 0$ and L' Hospital's rule. When

$\bar{z}_u = 1$ and $\bar{z}_l = 0$ for the case of full screen, Eq. (14h) gives $\varepsilon_1 = 0$ for $\beta_n > 0$ and the
second RHS terms of Eqs. (16a) – (16c) can therefore be eliminated. This causes the solution
for confined aquifers is independent of dimensionless elevation $\bar{z}$, indicating only horizontal
flow in the aquifer.

### 2.5. Pseudo-steady state solution for unconfined aquifer

A pseudo-steady state (PSS) solution $\bar{h}_s$ accounts for SHM of head fluctuation at a late period
of pumping time and satisfies the following form (Dagan and Rabinovich, 2014)
$\bar{h}_s(\bar{r}, \bar{z}, \bar{t}) = \text{Im}\big(\bar{H}(\bar{r}, \bar{z}) \, e^{i\gamma\bar{t}}\big)$          (17)
where $\bar{H}(\bar{r}, \bar{z})$ is a space function of $\bar{r}$ and $\bar{z}$. Define a PSS IGD model as Eqs. (8) - (13)
excluding (9), (11b) and replacing $\sin(\gamma\bar{t})$ in (10) by $e^{i\gamma\bar{t}}$. Substituting Eq. (17) and
$\partial\bar{h}_s/\partial\bar{t} = \text{Im}\big(i\gamma\bar{H}(\bar{r}, \bar{z}) \, e^{i\gamma\bar{t}}\big)$ into the model results in
$\frac{\partial^2 \bar{H}}{\partial\bar{r}^2} + \frac{1}{\bar{r}} \frac{\partial\bar{H}}{\partial\bar{r}} + \mu \frac{\partial^2 \bar{H}}{\partial\bar{z}^2} = i\gamma\bar{H}$          (18)
$\frac{\partial\bar{H}}{\partial\bar{r}} = \begin{cases} 1 & \text{for } \bar{z}_l \leq \bar{z} \leq \bar{z}_u \\ 0 & \text{outside screen interval} \end{cases}$ at $\bar{r} = 1$          (19)
$\frac{\partial\bar{H}}{\partial\bar{z}} = -ia\gamma\bar{H}$ at $\bar{z} = 1$ for IGD          (20)
$\frac{\partial\bar{H}}{\partial\bar{z}} = 0$ at $\bar{z} = 0$          (21)
$\lim_{\bar{r}\to\infty} \bar{H} = 0$          (22)

The resultant model is independent of $\bar{t}$, indicating the analytical solution of $\bar{H}(\bar{r}, \bar{z})$ is

tractable. Similarly, consider a PSS DGD model that equals the PSS IGD model but replaces
(11a) by (11b). Substituting Eq. (17) into the result yields a model that depends on $\bar{t}$, indicating
the solution $\bar{h}_s$ to the PSS DGD model is not tractable.

The Weber transform, defined in Eq. (B.1) of the supporting material, may be considered

as a Hankel transform with a more general kernel function. It can be applied to diffusion-type
problems with a radial-symmetric region from a finite distance to infinity. For groundwater
flow problems, it can be used to develop the analytical solution for the flow equation with a
Neumann boundary condition specified at the rim of a finite-radius well (e.g., Lin and Yeh,
2017; Povstenko, 2015). Taking the transform and the formula of $e^{i\gamma\bar{t}} = \cos(\gamma\bar{t}) + i\sin(\gamma\bar{t})$
to solve Eqs. (18) - (22) yields the solution of $\bar{h}_s$ expressed as
$$\bar{h}_s(\bar{r}, \bar{z}, \bar{t}) = \bar{A}_s(\bar{r}, \bar{z})\cos(\gamma t - \phi_s(\bar{r}, \bar{z})) \tag{23a}$$
$$\bar{A}_s(\bar{r}, \bar{z}) = \sqrt{a_s(\bar{r}, \bar{z})^2 + b_s(\bar{r}, \bar{z})^2} \tag{23b}$$
$$a_s(\bar{r}, \bar{z}) = \mathrm{Re}(\bar{H}(\bar{r}, \bar{z})) \tag{23c}$$
$$b_s(\bar{r}, \bar{z}) = \mathrm{Im}(\bar{H}(\bar{r}, \bar{z})) \tag{23d}$$
$$\phi_s(\bar{r}, \bar{z}) = \cos^{-1}(b_s(\bar{r}, \bar{z})/A_s(\bar{r}, \bar{z})) \tag{23e}$$
$$\bar{H}(\bar{r}, \bar{z}) = \begin{cases} \int_0^\infty \tilde{H}_u\, \xi\, \Omega\, d\xi & \text{for } \bar{z}_u < \bar{z} \leq 1 \\ \int_0^\infty \tilde{H}_m\, \xi\, \Omega\, d\xi & \text{for } \bar{z}_l \leq \bar{z} \leq \bar{z}_u \\ \int_0^\infty \tilde{H}_l\, \xi\, \Omega\, d\xi & \text{for } 0 \leq \bar{z} < \bar{z}_l \end{cases} \tag{23f}$$
$$\Omega = (J_0(\xi\bar{r})Y_1(\xi) - Y_0(\xi\bar{r})J_1(\xi))/(J_1^2(\xi) + Y_1^2(\xi)) \tag{23g}$$
with the Bessel functions of the first kind of order zero $J_0(-)$ and one $J_1(-)$ as well as the
second kind of order zero $Y_0(-)$ and one $Y_1(-)$,
$$\begin{cases} \tilde{H}_u = \tilde{H}_p(c_1\exp(\lambda_w\bar{z}) + c_2\exp(-\lambda_w\bar{z})) & \text{for } \bar{z}_u < \bar{z} \leq 1 \\ \tilde{H}_m = \tilde{H}_p(c_3\exp(\lambda_w\bar{z}) + c_4\exp(-\lambda_w\bar{z}) - 1) & \text{for } \bar{z}_l \leq \bar{z} \leq \bar{z}_u \\ \tilde{H}_l = \tilde{H}_p c_5(\exp(\lambda_w\bar{z}) + \exp(-\lambda_w\bar{z})) & \text{for } 0 \leq \bar{z} < \bar{z}_l \end{cases} \tag{23h}$$
$$c_1 = -e^{-\lambda_w}(\lambda_w - \sigma)(\sinh(\bar{z}_l\lambda_w) - \sinh(\bar{z}_u\lambda_w))/D \tag{23i}$$
$$c_2 = -e^{\lambda_w}(\lambda_w + \sigma)(\sinh(\bar{z}_l\lambda_w) - \sinh(\bar{z}_u\lambda_w))/D \tag{23j}$$
$$c_3 = \frac{e^{-(1+\bar{z}_l+\bar{z}_u)\lambda_w}}{2D}(\sigma(e^{(2+\bar{z}_l)\lambda_w} + e^{\bar{z}_u\lambda_w} - e^{(2\bar{z}_l+\bar{z}_u)\lambda_w}) + (\sigma - \lambda_w)e^{(\bar{z}_l+2\bar{z}_u)\lambda_w} +$$
$$\lambda_w(e^{(2+\bar{z}_l)\lambda_w} - e^{\bar{z}_u\lambda_w} + e^{(2\bar{z}_l+\bar{z}_u)\lambda_w})) \tag{23k}$$
$$c_4 = \frac{e^{-(1+\bar{z}_l+\bar{z}_u)\lambda_w}}{2D}((\sigma - \lambda_w)e^{(\bar{z}_l+2\bar{z}_u)\lambda_w} + (\sigma + \lambda_w)(e^{(2+\bar{z}_l)\lambda_w} - e^{(2+\bar{z}_u)\lambda_w} +$$
$e^{(2+2\bar{z}_l+\bar{z}_u)\lambda_w)})$ (23l)
$c_5 = \frac{1}{2D} e^{-(1+\bar{z}_l+\bar{z}_u)\lambda_w} (e^{\bar{z}_l\lambda_w} - e^{\bar{z}_u\lambda_w})((\lambda_w - \sigma)e^{(\bar{z}_l+\bar{z}_u)\lambda_w} + (\lambda_w + \sigma)e^{2\lambda_w})$ (23m)
where $\lambda_w^2 = (\xi^2 + i\gamma)/\mu$ , $\sigma = i\gamma a$ , $\tilde{H}_p = 2/(\pi\mu\xi\lambda_w^2)$ and $D = 2(\sigma \cosh \lambda_w +$
$\lambda_w \sinh \lambda_w)$, and Re(-) is the real part of a complex number. Again, one can refer to the
supporting material for the derivation of the solution. Eq. (23a) indicates SHM for the response
of the hydraulic head at any point to oscillatory pumping. Note that Eq. (23f) reduces to
$\bar{H}(\bar{r},\bar{z}) = \int_0^\infty \tilde{H}_m \xi \Omega \, d\xi$ for a fully screened well when $\bar{z}_l = 0$ and $\bar{z}_u = 1$.
**2.6. Pseudo-steady state solution for confined aquifers**
Applying the finite Fourier cosine transform to the model, Eqs. (18) – (22) with $S_y = 0$ (i.e.,
$a = 0$) for the confined condition, leads to an ordinary differential equation with two boundary
conditions. With solving the boundary-value problem, the solution of $\bar{h}_s$ for confined aquifers
can be expressed as Eqs. (23a) - (23e) with $\bar{H}(\bar{r},\bar{z})$ defined as
$\bar{H}(\bar{r},\bar{z}) = -2\sum_{m=0}^\infty \frac{K_0(\bar{r}\lambda_m)}{\lambda_m K_1(\lambda_m)} \times \begin{cases} 0.5(\bar{z}_u - \bar{z}_l) & \text{for } m = 0 \\ \frac{\cos(m\pi\bar{z})}{m\pi}(\sin(\bar{z}_u m\pi) - \sin(\bar{z}_l m\pi)) & \text{for } m > 0 \end{cases}$ (24)
where $\lambda_m^2 = \gamma i + \mu(m\pi)^2$. The derivation of Eq. (24) is also listed in the supporting material.
For a fully screened well (i.e., $\bar{z}_u = 1$, $\bar{z}_l = 0$), the first term of the series (i.e., $m = 0$) remains
and the others equal zero because of $\sin(\bar{z}_u m\pi) - \sin(\bar{z}_l m\pi) = 0$. The result is independent
of dimensionless elevation $\bar{z}$, indicating the confined flow is only horizontal.
**2.7. Special cases of the present solution**
Table 1 classifies the present solution (i.e., Solution 1) and its special cases (i.e., Solutions 2 to
6) according to transient or PSS flow, unconfined or confined aquifer, and IGD or DGD. Each
of Solutions 1 to 6 reduces to a special case for fully screened well. Existing analytical solutions
can be regarded as special cases of the present solution as discussed in Section 3.4 (e.g., Black
and Kipp, 1981; Rasmussen et al., 2003; Dagan and Rabinovich, 2014).
**2.8. Sensitivity analysis**
Sensitivity analysis evaluates hydraulic head variation in response to the change in each of $K_r$,
$K_z$, $S_s$, $S_y$, $\omega$, and $\varepsilon$. The normalized sensitivity coefficient can be defined as (Liou and Yeh,

1997)

$S_i = P_i \dfrac{\partial X}{\partial P_i}$                                                                          (25)
where $S_i$ is the sensitivity coefficient of $i$th parameter; $P_i$ is the magnitude of the $i$th input
parameter; $X$ represents the present solution in dimensional form. Eq. (25) can be approximated
as
$S_i = P_i \dfrac{X(P_i + \Delta P_i) - X(P_i)}{\Delta P_i}$                                                          (26)
where $\Delta P_i$, a small increment, is chosen as $10^{-3} P_i$.

## 3. Results and Discussion

The following sections demonstrate the response of the hydraulic head to oscillatory pumping
using the present solution. The default values in calculation are $r = 0.05$ m, $z = 5$ m, $b = 10$ m,
$Q = 10^{-3}$ m$^3$/s, $r_w = 0.05$ m, $z_u = 5.5$ m, $z_l = 4.5$ m, $K_r = 10^{-4}$ m/s, $K_z = 10^{-5}$ m/s, $S_s = 10^{-5}$ m$^{-1}$, $S_y$
$= 10^{-4}$, $\omega = 2\pi/30$ s$^{-1}$, and $\kappa = 100$ s. The corresponding dimensionless parameters and
variables are $\bar{r} = 1$, $\bar{z} = 0.5$, $\bar{z}_u = 0.55$, $\bar{z}_l = 0.45$, $\gamma = 5.24 \times 10^{-5}$, $\mu = 2.5 \times 10^{-6}$, $a =$
$4 \times 10^{5}$, $a_1 = 1$ and $a_2 = 2.5 \times 10^{-6}$.

### 3.1. Delayed gravity drainage

Previous analytical models for OPT consider either confined flow (e.g., Rasmussen et al.,

2003) or unconfined flow with IGD effect (e.g., Dagan and Rabinovich, 2014). Little attention
has been paid to the consideration of the DGD effect. This section addresses the diffrence
among these three models. Figure 2 shows the curve of the dimensionless amplitude $\bar{A}_t$ at $(\bar{r},$
$\bar{z}) = (1, 1)$ of Solution 1 versus the dimensionless parameter $a_1$ related to the DGD effect. The
transient head fluctuations are plotted based on Solution 1 with $a_1 = 10^{-2}$, 1, 10, 500,
Solution 2 for IGD and Solution 3 for confined flow. Define the relative error as
$RE = |\bar{A}'_t - \bar{A}_t| / \bar{A}_t$                                                                          (27)
where $\bar{A}'_t$ is the dimensionless amplitude predicted by Solution 2 for the case of $a_1 = 500$
or Solution 3 for the case of $a_1 = 10^{-2}$. The curves of the $RE$ versus the period of oscillatory
pumping rate (i.e., $P$) for these two cases are displayed. The range of $P \leq 10^5 s$ (1.16 d)
contains most practical applications of OPT. When $10^{-2} \leq a_1 \leq 500$, the $\bar{A}_t$ gradually
decreases with $a_1$ to the trough and then increases to the ultimate value of $\bar{A}_t = 1.79 \times 10^{-2}$.
The DGD, in other words, causes an effect. When $a_1 < 10^{-2}$, Solutions 1 and 3 agree on the
predicted heads; the $RE$ is below 1% for $P < 10^4 s$ (2.78 h), indicating the unconfined aquifer
with the DGD effect behaves like confined aquifer and the water table can be regarded as a no-
flow boundary when $a_1 < 10^{-2}$ and $P < 10^4 s$. When $a_1 > 500$, the head fluctuations
predicted by both Solutions 1 and 2 are identical; the largest $RE$ is about 0.45%, indicating the
DGD effect is ignorable and Eq. (4b) reduces to (4a) for the IGD condition. This conclusion is
applicable for any magnitude of $P$ in spite of $P > 10^5 s$.
**3.2. Effect of finite radius of pumping well**
Existing analytical models for OPT mostly treated the pumping well as a line source with
infinitesimal radius (e.g., Rasmussen et al., 2003; Dagan and Rabinovich, 2014). The finite
difference scheme for the model also treats the well as a nodal point by neglecting the radius.
These will lead to significant error when a well has the radius ranging from 0.5 m to 2 m (Yeh
and Chang, 2013). This section discusses the relative error in predicted amplitude defined as
$RE = |\bar{A}_{D\&R} - \bar{A}_t|/\bar{A}_t$                    (28)
where $\bar{A}_t$ and $\bar{A}_{D\&R}$ are the dimensionless amplitudes at $\bar{r} = 1$ (i.e., $r = r_w$) predicted by IGD
Solution 2 and the Dagan and Rabinovich (2014) solution, respectively. Note that their solution
assumes infinitesimal radius of a pumping well and has a typo that the term $e^{-D_w+1} - e^{-D_w}$
should read $e^{\beta(-D_w+1)} - e^{-\beta D_w}$ (see their Eq. (25)). Figure 3 demonstrates the $RE$ for
different values of radius $r_w$. The RE increases with $r_w$ as expected. For case 1 of $r_w = 0.1$ m,
both solutions agree well in the entire domain of $1 \leq \bar{r} \leq \infty$, indicating a pumping well with
$r_w \leq 0.1$ m can be regarded as a line source. For the extreme case 2 of $r_w = 1$ m or case 3 of
$r_w = 2$ m, the Dagan and Rabinovich solution underestimates the dimensionless amplitude for
$1 \leq \bar{r} \leq 6$ and agrees to the present solution for $\bar{r} > 6$. The $RE$s for these two cases exceed
10%. The effect of finite radius should therefore be considered in OPT models especially when
observed hydrulic head data are taken close to the wellbore of a large-diameter well.

### 3.3. Sensitivity analysis

The temporal distributions of normalized sensitivity coefficient $S_i$ defined as Eq. (26) with
$X = h_{\exp}$ of Solution 1 are displayed in Fig. 4a for the response of exponential decay to the
change in each of six parameters $K_r$, $K_z$, $S_s$, $S_y$, $\omega$ and $\varepsilon$. The exponential decay is very sensitive
to variation in each of $K_r$, $K_z$, $S_s$ and $\omega$ because of $|S_i| > 0$. Precisely, a positive perturbation
in $S_s$ produces an increase in the magnitude of $h_{\exp}$ while that in $K_r$ or $K_z$ causes a decrease.
In addition, a positive perturbation in $\omega$ yields an increase in $h_{\exp}$ before $t = 1$ s and a decrease
after that time. It is worth noting that $S_i$ for $S_y$ or $\varepsilon$ is very close to zero over the entire period
of time, indicating $h_{\exp}$ is insensitive to the change in $S_y$ or $\varepsilon$ and the subtle change of gravity
drainage has no influence on the exponential decay. On the other hand, the spatial distributions
of $S_i$ associated with the amplitude $A_t$ are shown in Fig. 4b in response to the changes in
those six parameters. The $A_t$ is again sensitive to the change in each of $K_r$, $K_z$, $S_s$ and $\omega$ but
insensitive with the change in $S_y$ or $\varepsilon$. The same result of $|S_i| \cong 0$ for $S_y$ or $\varepsilon$ applies to any
observation point under the water table (i.e., $\bar{z} < 1$), but $|S_i| > 0$ at the water table (i.e., $\bar{z} =$
1) shown in Fig. 4c. From those discussed above, we may conclude the changes in the four key
parameters $K_r$, $K_z$, $S_s$ and $\omega$ significantly affect head prediction in the entire aquifer domain.
The change in $S_y$ or $\varepsilon$ leads to insignificant variation in the predicted head below the water
table and slight variation at the water table.

### 3.4. Transient head fluctuation affected by the initial condition

Figure 5 demonstrates head fluctuations predicted by DGD Solution 1 and IGD Solution 2
expressed as $\bar{h} = \bar{h}_{\exp} + \bar{h}_{\mathrm{SHM}}$ for transient flow and by IGD solution as $\bar{h}_s = \bar{A}_s \cos(\gamma t -$
$\phi_s)$ for PSS flow. The transient head fluctuation starts from $\bar{h} = 0$ at $\bar{t} = 0$ and approaches
SHM predicted by $\bar{h}_{SHM}$ when $\bar{h}_{exp} \cong 0$ m after $\bar{t} = 0.5\bar{P}$ (i.e., $6 \times 10^4$). Solutions 1 and
2 agree to the $\bar{h}$ predictions because the head at $\bar{z} = 0.5$ under the water table is insensitive
to the change in $S_y$ or $\varepsilon$ as discussed in Section 3.3. It is worth noting that the solution of
Dagan and Rabinovich (2014) for PSS flow has a time shift from the $\bar{h}_{SHM}$ of Solution 2. This
indicates the phase of their solution (i.e., 1.50 rad) should be replaced by the phase of Solution
2 (i.e., $\phi_t = 1.64$ rad) so that their solution exactly fits the $\bar{h}_{SHM}$ of Solution 2.

Figure 6 displays head fluctuations predicted by transient Solution 3 expressed as $\bar{h} =$

$\bar{h}_{exp} + \bar{h}_{SHM}$ and PSS Solution 6 as $\bar{h}_s = \bar{A}_s \cos(\gamma t - \phi_s)$ for partially screened pumping
well in panel (a) and full screen in panel (b). The Rasmussen et al. (2003) solution for transient
flow predicts the same $\bar{h}$ as Solution 3. The Black and Kipp (1981) for PPS flow also predicts
close $\bar{h}_{SHM}$ prediction of Solution 3. The phase of Solution 6 (i.e., $\phi_s = 1.50$ rad for panel (a)
and 1.33 rad for (b)) can be replaced by the phase of Solution 3 (i.e., $\phi_t = 1.64$ rad for (a)
and 1.81 rad for (b)) so that the $\bar{h}_{SHM}$ prediction of Solutions 3 is identical to the $\bar{h}_s$
prediction of Solution 6. As concluded, excluding the initial condition with Eq. (17) for a PSS
model leads to a time shift from the SHM of the head fluctuation predicted by the associated
transient model while the transient and PSS models give the same SHM amplitude.
**3.5. Application of the present solution to field experiment**
Rasmussen et al. (2003) conducted field OPTs in a three-layered aquifer system containing one
Surficial Aquifer, the Barnwell-McBean Aquifer in between and the deepest Gordon Aquifer
at the Savannah River site. Two clay layers dividing these three aquifers may be regarded as
impervious strata. For the OPT at the Surficial Aquifer, the formation has 6.25 m averaged
thickness near the test site. The fully-screened pumping well has 7.6 cm outer radius. The
pumping rate can be approximated as $Q\sin(\omega t)$ with $Q = 4.16 \times 10^{-4}$ m$^3$/s and $\omega = 2\pi$ h$^{-1}$. The
distance from the pumping well is 6 m to the observation well 101D and 11.5 m to well 102D.
The screen lengths are 3 m from the aquifer bottom for well 101D and from the water table for
well 102D. For the OPT at the Barnwell-McBean Aquifer, the formation mainly consists of
sand and fine-grained material. The pumping well has outer radius of 7.6 cm and pumping rate
of $Q\sin(\omega t)$ with $Q = 1.19 \times 10^{-3}$ m$^3$/s and $\omega = \pi$ h$^{-1}$. The observation well 201C is at 6 m
from the pumping well. The data of time-varying hydraulic heads at the observation wells (i.e.,
101D, 102D, 201C) are plotted in Fig. 7. One can refer to Rasmussen et al. (2003) for detailed
description of the Savannah River site.
The aquifer hydraulic parameters are determined based on Solutions 3 to 6 coupled with
the Levenberg–Marquardt algorithm provided in the Mathematica function FindFit (Wolfram,
1991). Note that a robust Gauss-Newton algorithm provides an alternative for the parameter
estimation (Qin et al., 2018a; 2018b). Solutions 4 and 5 are used to predict depth-averaged
head expressed as $(z_u' - z_l')^{-1} \int_{z_l'}^{z_u'} h_s dz$ with the upper elevation $z_u'$ and lower one $z_l'$ of
the finite screen of the observation well 101D or 102D at the Surficial Aquifer. Note that
Solutions 3 and 6 are independent of elevation because of the fully-screened pumping well.
Define the standard error of estimate (SEE) as $\text{SEE} = \sqrt{\frac{1}{M}\sum_{j=1}^{M} e_j^2}$ and the mean error (ME)
as $\text{ME} = \frac{1}{M}\sum_{j=1}^{M} e_j$ where $e_j$ is the difference between predicted and observed hydraulic heads
and $M$ is the number of observation data (Yeh, 1987). The estimated parameters and associated
SEE and ME are displayed in Table 2. The estimates of $T$, $S$ and $D_r$ given in Rasmussen et al.
(2003) are also presented. The result shows the estimated $S_y$ is very small, and the estimated $T$
and $S$ by Solution 3, 6 or the Rasmussen et al. (2003) solution for confined flow are close to
those by Solution 4 or 5 for unconfined flow, indicating that the unconfined flow induced by
the OPT in the Surficial Aquifer is negligibly small. Little gravity drainage due to the DGD
effect appears with $a_1 = 20$ for wells 101D and 102D as discussed in Section 3.1. Rasmussen
et al. (2003) also revealed the confined behaviour of the OPT-induced flow in the Surficial
Aquifer. The estimated $S_y$ is one order less than the lower limit of the typical range of 0.01 ~
0.3 (Freeze and Cherry, 1979), which accords with the findings of Rasmussen et al. (2003) and
Rabinovich et al. (2015). Such a fact might be attributed to the problem of the moisture
exchange limited by capillary fringe between the zones below and above the water table.
Several laboratory research outcomes have confirmed an estimate of $S_y$ at short period of OPT
is much smaller than that determined by constant-rate pumping test (e.g., Cartwright et al.,
2003; 2005). In addition, the difference in $T$, $S$ or $D_r$ estimated by Solution 6 and those by the
Rasmussen et al. (2003) solution may be attributed to the fact that their solution assumes
isotropic hydraulic conductivity (i.e., $K_r = K_z$). On the other hand, transient Solution 3 gives
smaller SEEs than PSS Solution 6 or the Rasmussen et al. (2003) solution for the Barnwell-
McBean Aquifer and better fits to the observed data at the early pumping periods as shown in
Fig. 7. From those discussed above, we may conclude the present solution is applicable to real-
world OPT.

## 4. Concluding remarks

A variety of analytical models for OPT have been proposed so far, but little attention is paid to
the joint effects of DGD, initial condition, and finite radius of a pumping well. This study
develops a new model for describing hydraulic head fluctuation due to OPT in unconfined
aquifers. Static hydraulic head prior to OPT is regarded as an initial condition. A Neumann
boundary condition is specified at the rim of a finite-radius pumping well. A free surface
equation accounting for the DGD effect is considered as the top boundary condition. The
solution of the model is derived by the Laplace transform, finite-integral transform and Weber
transform. The sensitivity analysis of the head response to the change in each of hydraulic
parameters is performed. The observation data obtained from the OPT at the Savannah River
site are analyzed by the present solution when coupling the Levenberg–Marquardt algorithm
to estimate aquifer hydraulic parameters. Our findings are summarized below:
1. When $10^{-2} \leq a_1 \leq 500$, the effect of DGD on head fluctuations should be considered.

The amplitude of head fluctuation predicted by DGD Solution 1 decreases with increasing

$a_1$ to a trough and then increases to the amplitude predicted by IGD Solution 2. When

$a_1 > 500$, the DGD becomes IGD. Both Solutions 1 and 2 predict the same head

fluctuation. When $a_1 < 10^{-2}$ and $P < 10^4$s, the DGD results in the water table under
no-flow condition. Solution 1 for unconfined flow gives an identical head prediction to
Solution 3 for confined flow.
2.  Assuming a large-diameter well as a line source with infinitesimal radius underestimates

the amplitude of head fluctuation in the domain of $1 \leq \bar{r} \leq 6$ when the radius exceeds 80

424        cm, leading to relative error RE > 10% shown in Fig. 3. In contrast, the assumption is valid

in predicting the amplitude in the domain of $\bar{r} > 6$ in spite of adopting a large-diameter

well. When $r_w \leq 10$ cm (i.e., RE < 0.45%), the well radius can be regarded as

infinitesimal. The result is applicable to existing analytical solutions assuming infinitesimal

radius and finite difference solutions treating the pumping well as a nodal point.

3.  The sensitivity analysis suggests the changes in four parameters $K_r$, $K_z$, $S_s$ and $\omega$

significantly affect head prediction in the entire aquifer domain. The change in $S_y$ or $\varepsilon$

causes insignificant variation in the head under water table but slight variation at the water

table.

4.  Analytical solutions for OPT are generally expressed as the sum of the exponential and

harmonic functions of time (i.e., $\bar{h} = \bar{h}_{\exp} + \bar{A}_t \cos(\gamma t - \phi_t)$) for transient solutions (e.g.,

Solution 3) and harmonic function (i.e., $\bar{h}_s = \bar{A}_s \cos(\gamma t - \phi_s)$) for PSS solutions (e.g.,

Solution 6). The latter assuming Eq. (17) without the initial condition produces a time shift

from the SHM predicted by the $\bar{h}_{SHM}$. The phase $\phi_s$ should be replaced by $\phi_t$ so that

$\bar{h}_s$ and $\bar{h}_{SHM}$ are exactly the same.

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

**Acknowledgments**

Research leading to this paper has been partially supported by the grants from the Fundamental Research Funds for the Central Universities (2018B00114), the National Natural Science Foundation of China (51809080, 51879068, and 41561134016), National Key Research and Development Program (2018YFC0407900), and the Taiwan Ministry of Science and Technology under the contract numbers MOST 107-2221-E-009-019-MY3. The authors are grateful to Prof. T. C. Rasmussen for kindly providing the OPT data obtained from the Savannah River site.

**Table 1.** The present solution and its special cases

| Well screen | Transient flow | | Pseudo-steady state flow | |
|---|---|---|---|---|
| | Unconfined aquifer | Confined aquifer | Unconfined aquifer | Confined aquifer |
| Partial | Solutions 1 and 2 | Solution 3 | Solutions 4 and 5 | Solution 6 |
| Full | Solutions 1 and 2[a] | Solution 3[a,b] | Solutions 4 and 5[a] | Solution 6[a,b] |

Solution 1 consists of Eqs. (14a) – (14k) with the roots of Eq. (15) and $c_0 = a_1 p_0/(p_0 + a_2)$ for DGD.

Solution 2 is the same as Solution 1 but has $c_0 = a p_0$ for IGD.

Solution 3 equals Solution 1 with Eqs. (16a) – (16d) and $\beta_n = 0, \ \pi, \ 2\pi, \ \ldots, \ n\pi$.

Solution 4 is the component $\bar{h}_{\text{SHM}}$ of Solution 1 for DGD.

Solution 5 consists of Eqs. (23a) – (23m) for IGD.

Solution 6 consists of Eqs. (23a) – (23e) with $\bar{H}(\bar{r}, \bar{z})$ defined by Eq. (24).

[a] $\bar{z}_u = 1$ and $\bar{z}_l = 0$ for fully screened well

[b] The solution is independent of elevation.


**Table 2.** Hydraulic parameters estimated by the present solution and the Rasmussen et al. (2003) solution for OPT data from the Savannah River site

| Observation well | Solution | $T$ (m²/s) | $S$ | $D_r$ (m²/s) | $K_z$ (m/s) | $S_y$ | $C_y$ (m/s) | $\alpha$ | $\kappa$ (s) | SEE | ME |
|---|---|---|---|---|---|---|---|---|---|---|---|
| | | | | | *Surficial Aquifer* | | | | | | |
| 101D | Solution 3[a] | $9.27 \times 10^{-4}$ | $2.44 \times 10^{-3}$ | 0.380 | - | - | - | - | - | 0.018 | $-5.56 \times 10^{-3}$ |
| | Solution 6[b] | $9.18 \times 10^{-4}$ | $2.33 \times 10^{-3}$ | 0.393 | - | - | - | - | - | 0.018 | $-2.20 \times 10^{-4}$ |
| | Solution 4[c] | $4.61 \times 10^{-4}$ | $3.95 \times 10^{-3}$ | 0.117 | $7.38 \times 10^{-6}$ | $2.23 \times 10^{-3}$ | $3.31 \times 10^{-3}$ | 0.10 | 94.34 | 0.018 | $-2.20 \times 10^{-4}$ |
| | Solution 5[c] | $5.25 \times 10^{-4}$ | $1.09 \times 10^{-3}$ | 0.482 | $2.61 \times 10^{-5}$ | $5.49 \times 10^{-3}$ | $4.75 \times 10^{-3}$ | 0.31 | - | 0.019 | $-2.30 \times 10^{-4}$ |
| | Rasmussen et al. (2003)[b] | $2.17 \times 10^{-3}$ | $1.35 \times 10^{-4}$ | 16.074 | - | - | - | - | - | 0.018 | $-2.20 \times 10^{-4}$ |
| 102D | Solution 3[a] | $9.13 \times 10^{-4}$ | $1.76 \times 10^{-3}$ | 0.519 | - | - | - | - | - | 0.010 | $-4.38 \times 10^{-3}$ |
| | Solution 6[b] | $9.17 \times 10^{-4}$ | $1.67 \times 10^{-3}$ | 0.549 | - | - | - | - | - | 0.011 | $9.57 \times 10^{-4}$ |
| | Solution 4[c] | $9.57 \times 10^{-5}$ | $7.85 \times 10^{-4}$ | 0.122 | $3.68 \times 10^{-6}$ | $4.95 \times 10^{-3}$ | $7.43 \times 10^{-4}$ | 0.24 | 420.17 | 0.011 | $9.57 \times 10^{-4}$ |
| | Solution 5[c] | $9.49 \times 10^{-5}$ | $3.25 \times 10^{-4}$ | 0.292 | $4.67 \times 10^{-6}$ | $4.68 \times 10^{-3}$ | $9.98 \times 10^{-4}$ | 0.31 | - | 0.011 | $9.50 \times 10^{-4}$ |
| | Rasmussen et al. (2003)[b] | $2.27 \times 10^{-3}$ | $2.28 \times 10^{-4}$ | 9.956 | - | - | - | - | - | 0.011 | $9.57 \times 10^{-4}$ |
| | | | | | *Barnwell-McBean Aquifer* | | | | | | |
| 201C | Solution 3[a] | $5.86 \times 10^{-5}$ | $7.07 \times 10^{-4}$ | 0.083 | - | - | - | - | - | 0.232 | 0.046 |
| | Solution 6[b] | $6.03 \times 10^{-5}$ | $6.54 \times 10^{-4}$ | 0.092 | - | - | - | - | - | 0.363 | 0.281 |
| | Rasmussen et al. (2003)[b] | $6.90 \times 10^{-5}$ | $4.74 \times 10^{-4}$ | 0.150 | - | - | - | - | - | 0.363 | 0.281 |

[a] transient confined flow
[b] PSS confined flow
[c] PSS unconfined flow

# Figures

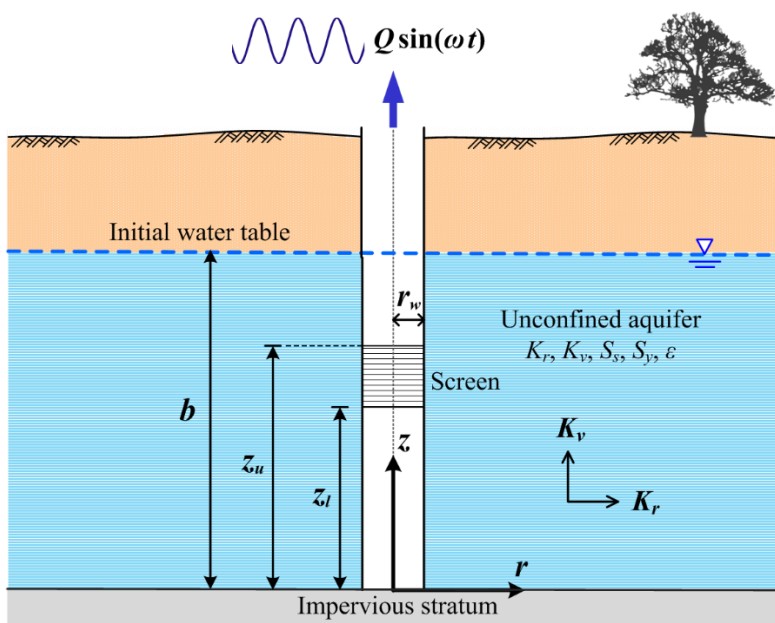


**Figure 1.** Schematic diagram for oscillatory pumping test at a partially screened well of finite
radius in an unconfined aquifer.

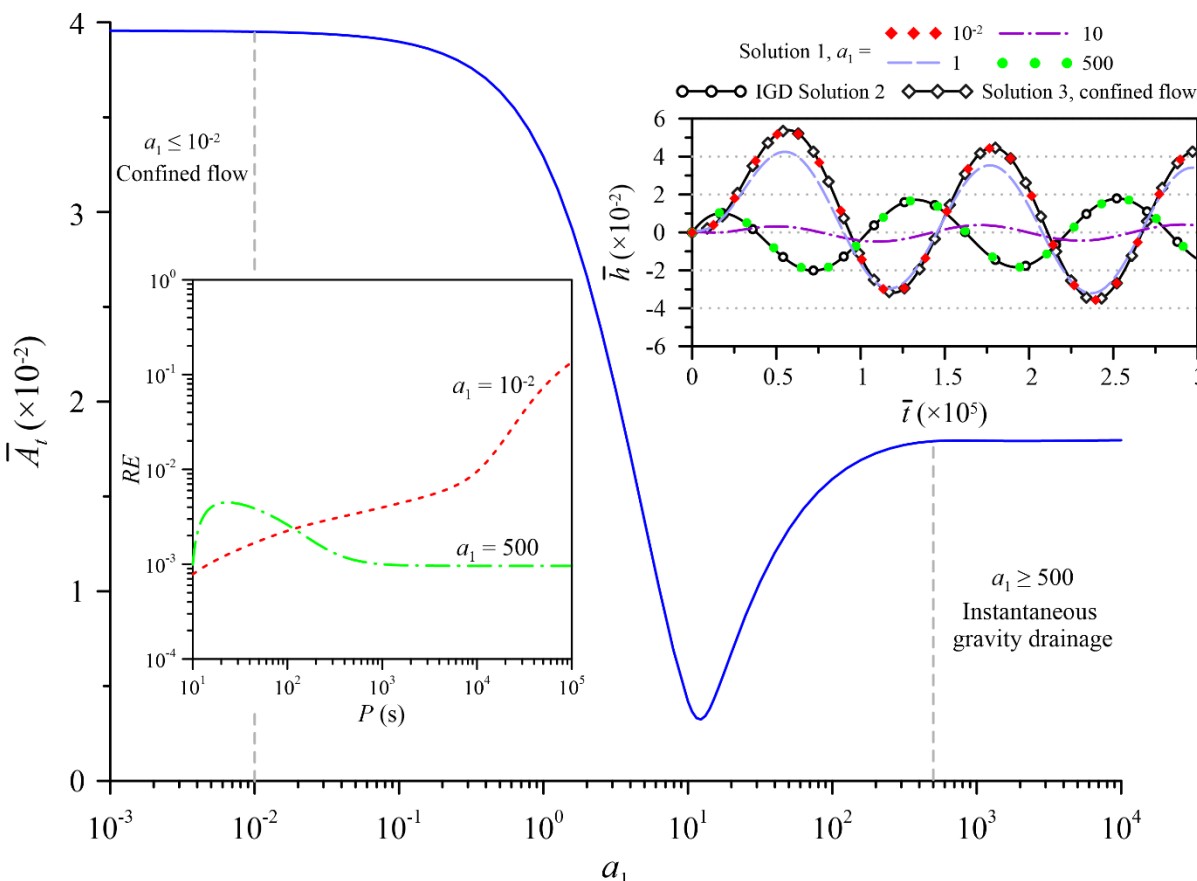


**Figure 2.** Influence of delayed gravity drainage on the dimensionless amplitude $\bar{A}_t$ and
transient head $\bar{h}$ at $\bar{r} = 1$, $\bar{z} = 1$ predicted by Solution 1 for different magnitudes of $a_1$
related to the influence.

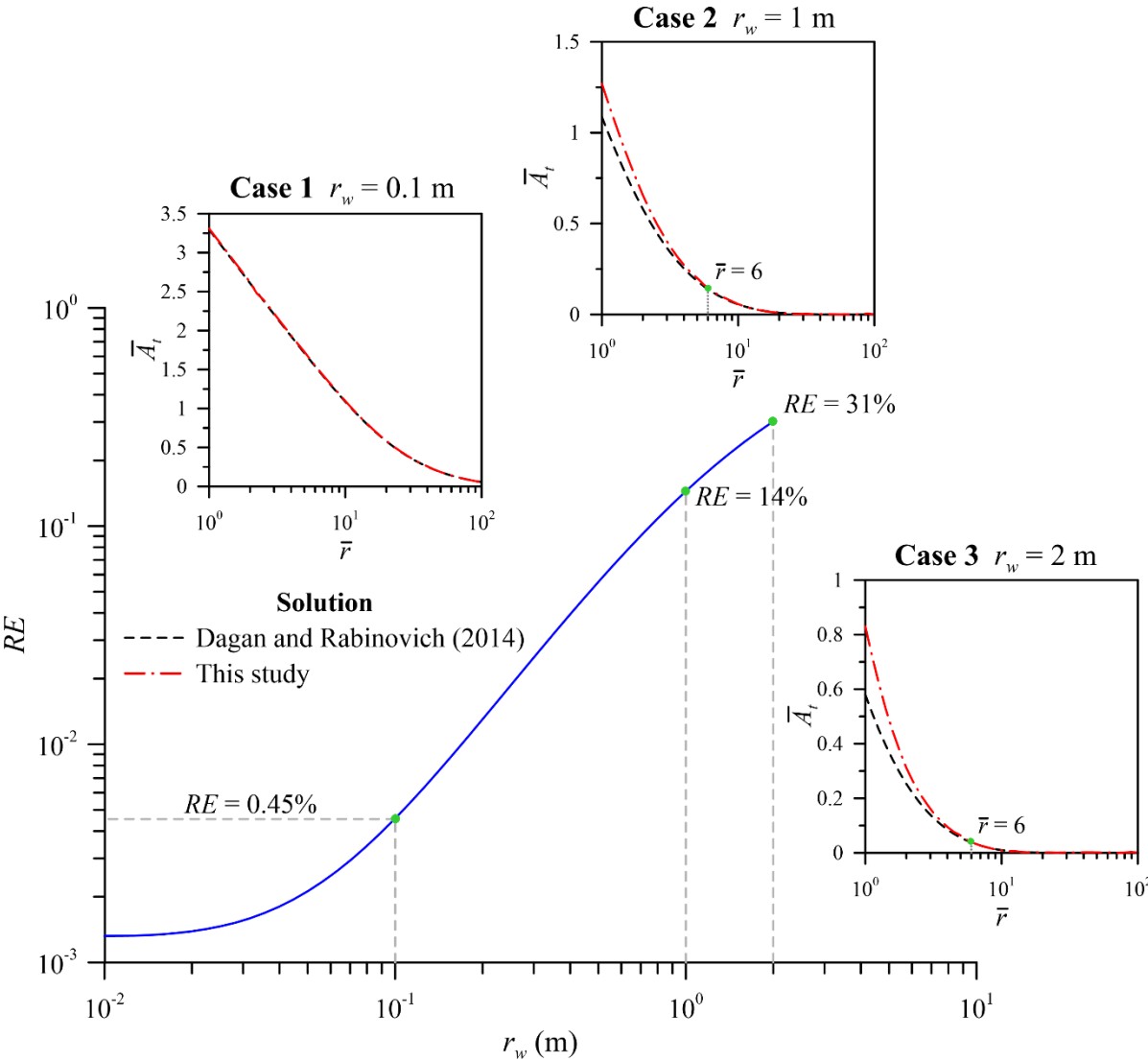


**Figure 3.** Relative error (*RE*) on the dimensionless amplitudes $\bar{A}_t$ at the rim of the pumping
well (i.e., $r = r_w$) predicted by IGD Solution 2 and the Dagan and Rabinovich (2014) solution.
The well radius is assumed infinitesimal in the Dagan and Rabinovich (2014) solution and
finite in our solution.

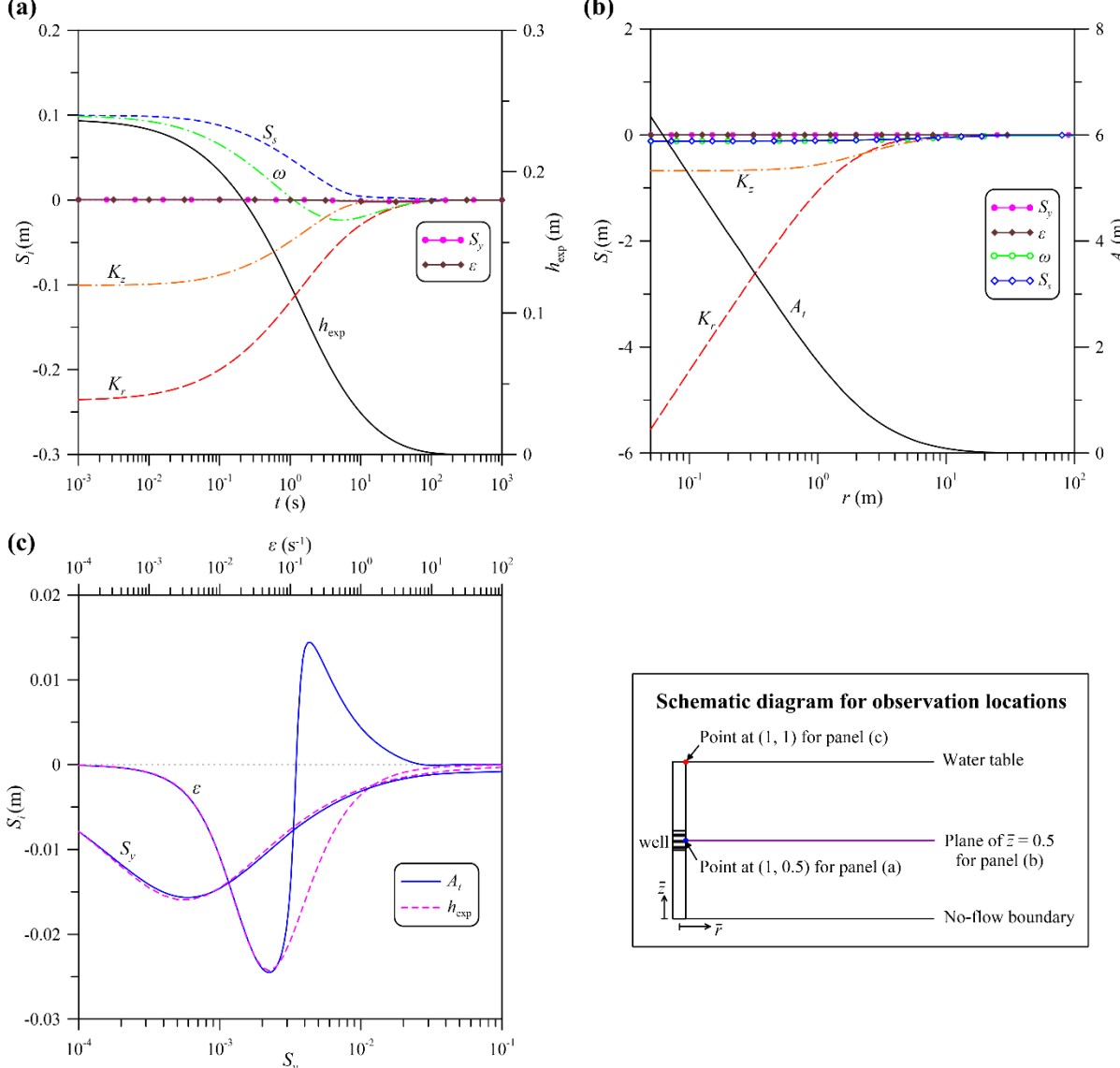

**Figure 4.** The normalized sensitivity coefficient $S_i$ associated with (a) the exponential component $h_{exp}$ of Solution 1 and (b) the SHM amplitude $A_t$ for parameters $K_r$, $K_z$, $S_s$, $S_y$, $\omega$ and $\varepsilon$. The observation locations for panels (a) and (b) are under water table (i.e., $\bar{z} = 0.5$). Panel (c) displays the curves of $S_i$ of $h_{exp}$ and $A_t$ at water table (i.e., $\bar{z} = 1$) versus $S_y$ and $\varepsilon$.

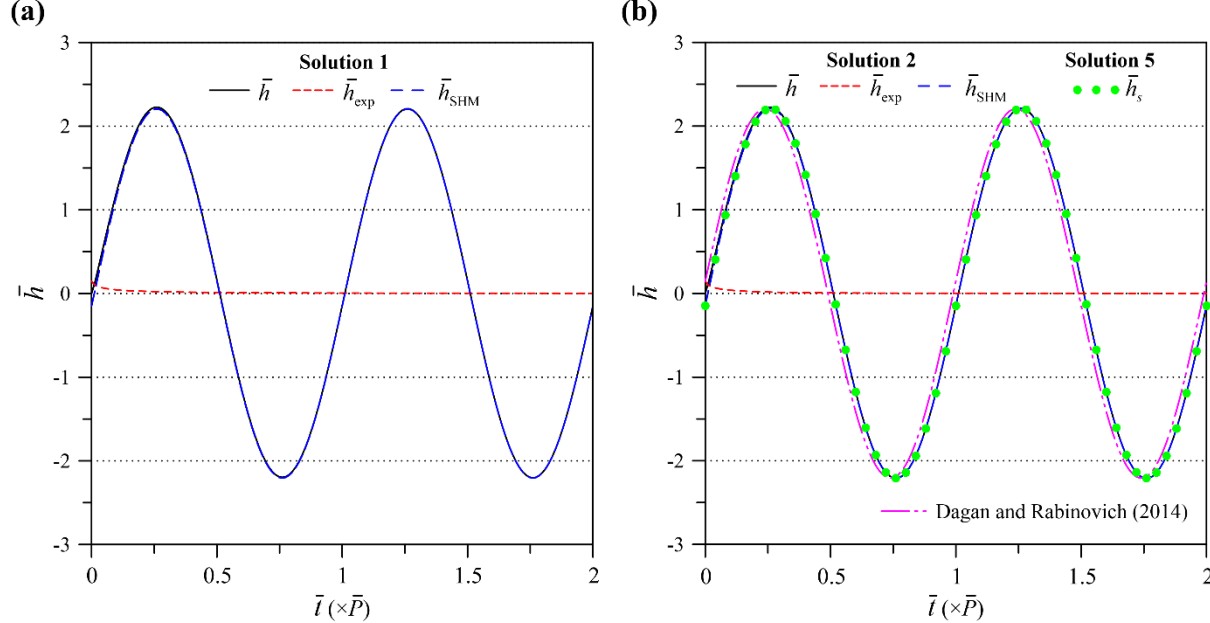

**Figure 5.** Heads fluctuations at $\bar{r} = 6$ predicted by (a) DGD Solution 1 and (b) IGD Solution 2. Solutions 1 and 2 are expressed as $\bar{h} = \bar{h}_{\text{exp}} + \bar{h}_{\text{SHM}}$ for transient flow. IGD Solution 5 expressed as $\bar{h}_s = \bar{A}_s \cos(\gamma t - \phi_s)$ accounts for PSS flow.

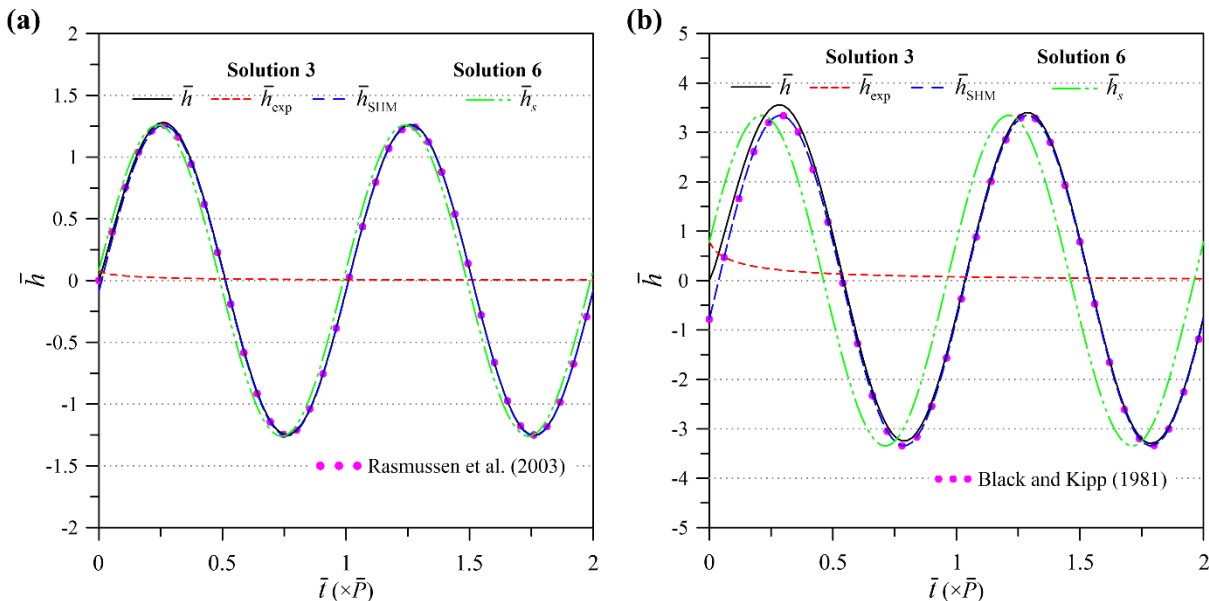


**Figure 6.** Heads fluctuations at $\bar{r} = 6$ predicted by Solutions 3 and 6 for (a) partially-screened
pumping well and (b) fully-screened pumping well. Solution 3 is expressed as $\bar{h} = \bar{h}_{exp} +$
$\bar{h}_{SHM}$ for transient flow. Solution 6 expressed as $\bar{h}_s = \bar{A}_s \cos(\gamma t - \phi_s)$ accounts for PSS
flow.

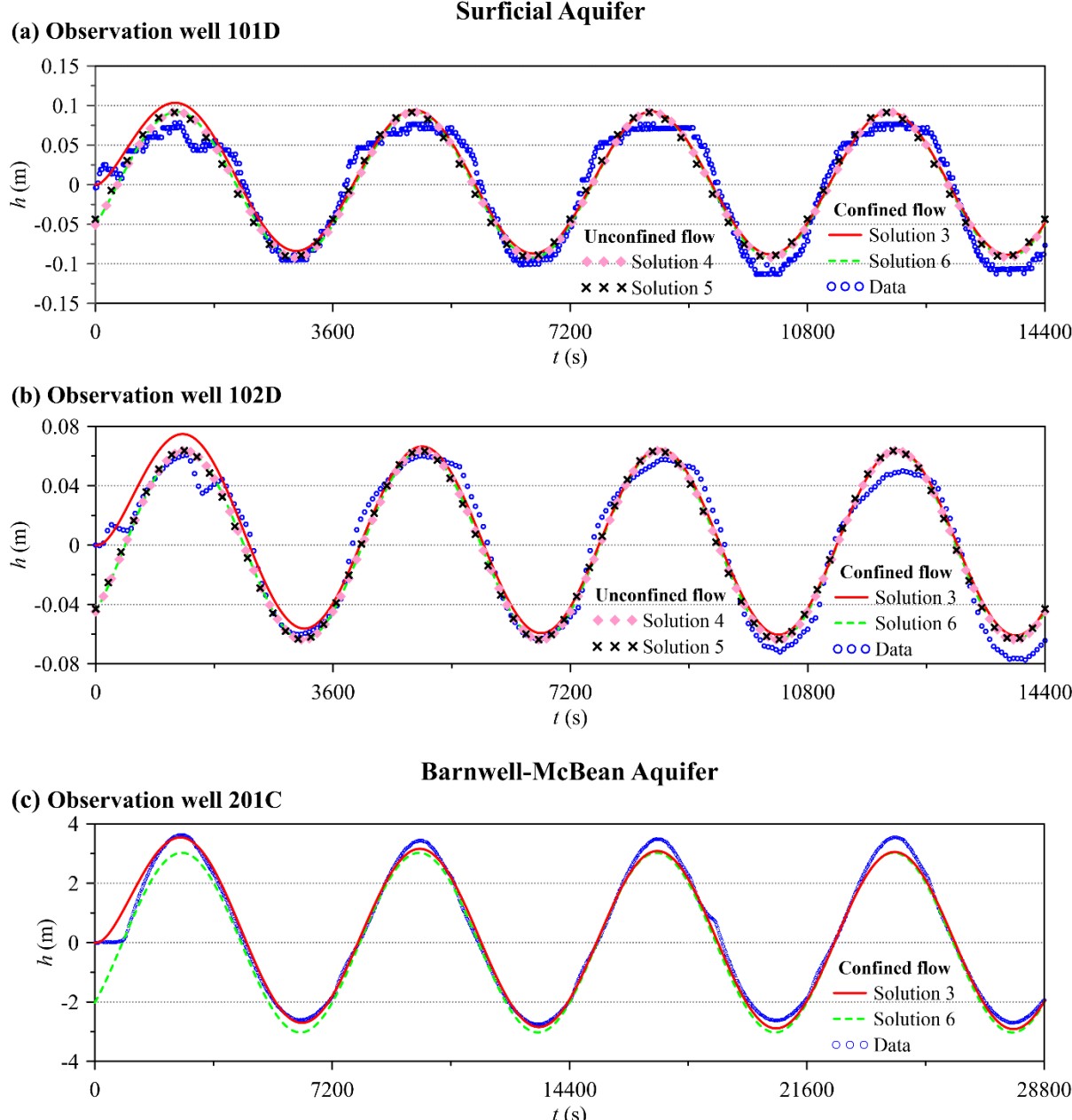


**Figure 7.** Comparision of field observation data with head fluctuations predicted by the present

solution. Solutions 3 and 6 represent transient and PSS confined flows, respectively. PSS

Solutions 4 and 5 stand for DGD and IGD conditions, respectively.
