# Peer review of "A General Analytical Model for Head Response to Oscillatory Pumping in"

_Hydrology and Earth System Sciences, 2018_

## Short Comment (SC1) · 5 Nov 2018

I miss a reference to: Huang, C.-S., Tsai, Y.-H., Yeh, H.-D., and Yang, T.: Analysis of Groundwater Response to Oscillatory Pumping Test in Unconfined Aquifers: Consider the Effects of Initial Condition and Wellbore Storage, Hydrol. Earth Syst. Sci. Discuss., https://doi.org/10.5194/hess-2018-199, 2018. which is closely related.

---

## Author Comment (AC1) · 5 Nov 2018

**Reply to Reviewer 1**

I miss a reference to: Huang, C.-S., Tsai, Y.-H., Yeh, H.-D., and Yang, T.: Analysis of Groundwater Response to Oscillatory Pumping Test in Unconfined Aquifers: Consider the Effects of Initial Condition and Wellbore Storage, Hydrol. Earth Syst. Sci. Discuss., https://doi.org/10.5194/hess-2018-199, 2018. which is closely related.

Response: This paper is actually a revised version because Huang et al. (2018) of the previous manuscript with no. hess-2018-199 was "rejected with invitation to resubmit". Frankly speaking, we think that previous reviewers' comments were full of personal prejudices. The previous manuscript, reviewers' comments, our replies, and editor's decision letter are available through the link (https://doi.org/10.5194/hess-2018-199).

Note that the present version has following two major changes:

(1) Prof. T. C. Rasmussen provided us raw data of hydraulic head fluctuation taken from field oscillatory pumping tests at the Savannah River site (Rasmussen et al., 2003). The data from the Boise Hydrogeophysical Research Site reported in Rabinovich et al. (2015) and its associated analyses had been removed.

(2) Different from the previous model, the present analytical model considers the effect of delayed gravity drainage (DGD) on water table motion in a new free surface equation. The present model can also deal with the case that the groundwater flow is subject to the effect of instantaneous gravity drainage (IGD). Head responses to the DGD and IGD effects are compared and discussed in the current version.

**References**

Huang, C.-S., Tsai, Y.-H., Yeh, H.-D., and Yang, T.: Analysis of Groundwater Response to Oscillatory Pumping Test in Unconfined Aquifers: Consider the Effects of Initial Condition and Wellbore Storage, Hydrol. Earth Syst. Sci. Discuss., https://doi.org/10.5194/hess-2018-199, 2018.

Rabinovich, A., Barrash, W., Cardiff, M., Hochstetler, D., Bakhos, T., Dagan, G., and Kitanidis, P. K.:

Frequency dependent hydraulic properties estimated from oscillatory pumping tests in an unconfined aquifer, J. Hydrol., 531, 2–16, 2015.

Rasmussen, T. C., Haborak, K. G., and Young, M. H.: Estimating aquifer hydraulic properties using sinusoidal pumping at the Savannah River site, South Carolina, USA, Hydrogeol. J., 11(4), 466–482, 2003.

---

## Referee Comment (RC1) · Anonymous Referee #1 · 20 Nov 2018

This paper represents a nice advancement of mathematical modeling of oscillatory pumping test.

I have the following comments on the paper:

1. Provide some more background on Weber transform and its application in hydrology, including, but not limited to its advantages and disadvantages.

2. A great portion of the mathematical details may be moved into supplementary material, so the authors can concentrate on discussing the hydrogeological features of the

problem.

3. The mathematical modeling appears to be robust. The English is good too.

4. Some associated literature using the similar approaches can be seen in Dr. Xiuyu Liang's recent publications (only one of them is cited here).

The paper can be published after moderate revision.

---

## Author Comment (AC2) · 3 Dec 2018

Please see the supplement.

Please also note the supplement to this comment:
https://www.hydrol-earth-syst-sci-discuss.net/hess-2018-482/hess-2018-482-AC2-supplement.zip

———————————————————

482, 2018.

---

## Referee Comment (RC2) · Rasmussen (Referee) · 8 Dec 2018

**Date:** December 8, 2018

**Review of:** "A General Analytical Model for Head Response to Oscillatory Pumping in Unconfined Aquifers: Consider the Effects of Delayed Gravity Drainage and Initial Condition" by HUANG Ching-Sheng, TSAI Ya-Hsin, YEH Hund-Der, and YANG Tao (HESS-2018-482)

**Review by:** Todd C Rasmussen, *trasmuss@uga.edu*

[Figure]

General Comments

1. This manuscript examines the response of water-table aquifers to periodic (sinusoidal, oscillatory) hydraulic perturbations.

   As noted in our periodic aquifer test at the Savannah River Site (Rasmussen et al 2003), the estimated storativity of the water-table aquifer more closely represented confined (early-time) as opposed to unconfined (late-time) conditions, and we speculated that the effects of delayed yield might explain this behavior.

   This manuscript examines this effect by comparing instantaneous and delayed yield solutions against each other as well as the observed field behavior. As such, it provides valuable new insight in the physics of water-table responses to hydraulic perturbations.

   Specifically, Section 3.5 is an accurate and thoughtful analysis of our (Rasmussen et al, 2003) periodic aquifer test at the Savannah River Site. This section is a valuable contribution showing the usefulness of the proposed technique.

2. The manuscript is well-written in clear and concise English. The tables and figures are also appropriate, clear, and well notated. I provide a few suggested edits as noted in a subsequent section.

3. Agree with Reviewer 1 that detailed mathematical derivation can be placed in an appendix.

4. Your model might be better formulated using alternative parameters (e.g., Depner and Rasmussen, 2017, *Hydrodynamics of Time-Periodic Groundwater Flow: Diffusion Waves in Porous Media*):

   (a) Equation 1 can be written more parsimoniously using:

   $$D_r \left[ \frac{\partial^2 h}{\partial r^2} + \frac{1}{r} \frac{\partial h}{\partial r} + \alpha \frac{\partial^2 h}{\partial z^2} \right] = \frac{\partial h}{\partial t} \qquad (1)$$

where $D_r = K_r/S_s$ and $\alpha = K_z/K_r$, which reduces the number of model parameters from three to two.

(b) Equation 4. The vertical flux at $z = b$ is:

$$q_z = -K_z \frac{\partial h}{\partial z}(4a)$$ (2)

which can be defined for DGD conditions using (Boulton, 1954):

$$q_z = \frac{S_y}{\kappa} \int_o^t \frac{\partial h}{\partial \tau} \exp \frac{-(t - \tau)}{\kappa} \, d\tau (4b)$$ (3)

where $\kappa = 1/\epsilon$, which has units of time rather than inverse time. Note that Eqn 4b reduces to IGD conditions as $\kappa \to \infty$:

$$q_z = S_y \frac{\partial h}{\partial t}(4c)$$ (4)

Solving for the boundary gradient gives:

$$\frac{\partial h}{\partial z} = -\frac{1}{\kappa \, C_y} \int_o^t \frac{\partial h}{\partial \tau} e^{-\frac{r-\tau}{\kappa}} \, d\tau (4d)$$ (5)

where $C_y = K_z/S_y$, with units of L/T.

(c) Note that $D_r$ and $\alpha$ are domain parameters defined by Eqn 1, $K_r$ is a boundary parameter defined by Eqn 3, and $C_y$ and $\kappa$ are boundary parameters defined by Eqn 4, where boundary parameters describe the aquifer characteristics on or near the boundary, and domain parameters describe the average characteristics within the interior of the aquifer. All other parameters (i.e., $K_z$, $S_s$, $S_y$) are hybrid domain-boundary parameters that are a composite of both boundary and domain characteristics.

(d) Dimensionless parameters in Eqn 7 can now be defined using:

$$\bar{t} = t \, (D_r/r_w^2) \quad \bar{P} = P \, (D_r/r_w^2) \quad \gamma = \omega \, (r_w^2/D_r) \quad \mu = \alpha \, (r_w^2/b^2) \quad a_1 = b/(\kappa \, C_y)$$ (6)

Suggested Edits

1. Title, suggest removing "Consider the"

2. Lines 22-24, suggest removing "without net water extraction" because a periodic test can be superimposed on a steady test.

   Also, "Oscillatory pumping tests (OPT) provide an alternative to constant-head and constant-rate tests for determining aquifer hydraulic parameters, with many analytical models available for parameter determination."

3. Lines 30-31, suggest revising to "The solution is derived using the Laplace, finite-integral, and Weber transforms."

4. Line 37, suggest explaining "certain time shift" here and subsequently.

5. Lines 56-58, suggest noting that periodic signals (depending on frequency) are likely to be observable at far greater distances than constant pumping because the signal-to-noise ratio for periodic testing is smaller due to the lack of noise at the testing frequency, unless there is interference from natural or artificial sources, such as solar and lunar periodicities.

6. Line 71, suggest explaining "certain period" here and subsequently.

7. Line 121, first reference to a partially penetrating pumping well; suggest highlighting in the abstract and introduction.

8. Line 165, suggest capitalizing "Section" here and subsequently (it's a proper noun).

9. Line 300, suggest capitalizing "Solution" here and subsequently (it's a proper noun).

10. Line 326, Figure 2 is the most interesting aspect of this manuscript; suggest explaining how period affects this plot. What happens when $P$ is longer or shorter than $\epsilon$ (or $\kappa = 1/\epsilon$, with units of time)? I suspect that a $P \gg \kappa$ will provide an estimate of $S_y$ (i.e., late-time), while $P \ll \kappa$ gives $S_s$ (early time). Is it possible to have a dimensionless ratio of $P/\kappa$?

11. Line 445, suggest explaining "certain trough".

12. Table 2, suggest providing estimated domain ($D_r$, $\alpha$), boundary ($K_r$, $C_y$, $\kappa$), and hybrid ($K_z$, $S_s$, $S_y$) parameters along with their individual standard errors. You might also provide the estimates from Rasmussen et al (2003).

Please also note the supplement to this comment:
https://www.hydrol-earth-syst-sci-discuss.net/hess-2018-482/hess-2018-482-RC2-supplement.pdf

---

## Referee Comment (RC3) · Anonymous Referee #3 · 16 Dec 2018

General comments:

Authors describe new analytical solutions to oscillatory pumping tests, applied to data collected at the Savanna River Site in South Carolina, USA, and published by Rasmussen et al. (2003). The solutions extend those published earlier by several authors, by now including delayed gravity drainage, finite radius pumping wells and initial conditions in the well bore. The solutions were well described and the writing was clear.

In general, I agree with the others reviewers that much of the in-depth derivations of the

solutions can be moved to the appendix or supplemental section so that the authors could spend more time on the geology and results of the study. As presented, only about 1.5 pages of the manuscript was devoted the testing of the solutions with real field data. Moving derivations to the appendix would also improve readability of the manuscript, which as presented is extremely dense and likely would appeal to a very few number of applied mathematicians and/or hydrologists. Simplify the presentation of the material, and more readers will take the time to read the manuscript, and cite the work.

Specific comments:

L340 – Yeh and Change (2013) not included in the references

L377 – replace 'to' with 'with'

L385 – check sentence that begins on this line; it is unclear as written and needs some clarification

L425 – replace 'researches' with 'research outcomes'

L443 – re-write portion of sentence as 'the effect of DGD on head fluctuations should be considered.'

---

## Referee Comment (RC4) · Rasmussen (Referee) · 24 Dec 2018

I have now read the revised manuscript and responses to review comments. The authors have assiduously incorporated my suggested revisions, and I believe the manuscript is suitable for publication in its present form.

Their submission provides a more robust approach for evaluating periodic testing in unconfined aquifers by extending parameter estimation to vertical anisotropy and delayed drainage. As such, it extends the utility and practicality of periodic aquifer testing.

---

## Author Comment (AC3) · 24 Dec 2018

Please see the supplement.

Please also note the supplement to this comment:
https://www.hydrol-earth-syst-sci-discuss.net/hess-2018-482/hess-2018-482-AC3-supplement.zip
* * *
[Figure]

482, 2018.

---

## Author Comment (AC4) · 24 Dec 2018

Please see the supplement.

Please also note the supplement to this comment:
https://www.hydrol-earth-syst-sci-discuss.net/hess-2018-482/hess-2018-482-AC4-supplement.zip
* * *
482, 2018.

---

## Author Comment (AC5) · 5 Jan 2019

I have now read the revised manuscript and responses to review comments. The authors have assiduously incorporated my suggested revisions, and I believe the manuscript is suitable for publication in its present form. Their submission provides a more robust approach for evaluating periodic testing in unconfined aquifers by extending parameter estimation to vertical anisotropy and delayed drainage. As such, it extends the utility and practicality of periodic aquifer testing.

Response: We very appreciate your time and efforts in providing valuable comments which do improve the quality and readability of our manuscript.

---

## Author Response (AR1)

**Reply to Reviewer 1**

This paper represents a nice advancement of mathematical modeling of oscillatory pumping test. I have the following comments on the paper:

 Provide some more background on Weber transform and its application in hydrology, including, but not limited to its advantages and disadvantages.

Response: Thanks for the comment. We add the following sentences in the revised manuscript. "The Weber transform, defined in Eq. (B.1) of the supporting material, may be considered as a Hankel transform with a more general kernel function. It can be applied to diffusion-type problems with a radial-symmetric region from a finite distance to infinity. For groundwater flow problems, it can be used to develop the analytical solution for the flow equation with a boundary condition of Dirichlet, Neumann, or Robin type specified at the rim of a finite-radius well (e.g., Lin and Yeh, 2017; Povstenko, 2015)." (Pages 10 - 11, lines 218 - 223)

2. A great portion of the mathematical details may be moved into supplementary material, so the authors can concentrate on discussing the hydrogeological features of the problem.

Response: Thanks for the suggestion. The derivation of the present solution has been moved to the supplementary material, and then the Methodology section is shortened. Please refer to the revised manuscript as attached.

3. The mathematical modeling appears to be robust. The English is good too.

**Response: Thanks.**

4. Some associated literature using the similar approaches can be seen in Dr. Xiuyu Liang's recent publications (only one of them is cited here).

Response: The reference of Liang et al. (2018) is added in line 149, page 8 of the revised manuscript, and the citations there are then changed to (Latinopoulos, 1985; Liang et al., 2017; 2018).

The paper can be published after moderate revision. Response: Many thanks.

**References:**

- Latinopoulos, P.: Analytical solutions for periodic well recharge in rectangular aquifers with 3rd-kind boundary-conditions, J. Hydrol., 77(1–4), 293–306, 1985.
- Liang, X., Zhan, H., Zhang, Y.-K., and Liu, J.: On the coupled unsaturated-saturated flow process induced by vertical, horizontal, and slant wells in unconfined aquifers, Hydrol. Earth Syst. Sci., 21,

1251–1262, 2017.

- Liang, X., Zhan, H., Zhang, Y.-K., Liu, J.: Underdamped slug tests with unsaturated-saturated flows by considering effects of wellbore skins, Hydrol. Process., 32, 968 980, 2018.
- Lin, Y.-C., Yeh, H.-D.: A lagging model for describing drawdown induced by a constant-rate pumping in a leaky confined aquifer, Water Resour. Res., 53, 8500 8511, 2017.
- Povstenko, Y.: Linear fractional diffusion-wave equation for scientists and engineers. New York, Birkhäser, 2015.

**Reply to Reviewer 2**

**Review of:** "A General Analytical Model for Head Response to Oscillatory Pumping in Unconfined Aquifers: Consider the Effects of Delayed Gravity Drainage and Initial Condition" by HUANG Ching-Sheng, TSAI Ya-Hsin, YEH Hund-Der, and YANG Tao (HESS-2018-482)

Review by: Todd C Rasmussen, trasmuss@uga.edu

**General Comments**

 This manuscript examines the response of water-table aquifers to periodic (sinusoidal, oscillatory) hydraulic perturbations. As noted in our periodic aquifer test at the Savannah River Site (Rasmussen et al 2003), the estimated storativity of the water-table aquifer more closely represented confined (early-time) as opposed to unconfined (late-time) conditions, and we speculated that the effects of delayed yield might explain this behavior.

This manuscript examines this effect by comparing instantaneous and delayed yield solutions against each other as well as the observed field behavior. As such, it provides valuable new insight in the physics of water-table responses to hydraulic perturbations.

Specifically, Section 3.5 is an accurate and thoughtful analysis of our (Rasmussen et al, 2003) periodic aquifer test at the Savannah River Site. This section is a valuable contribution showing the usefulness of the proposed technique.

2. The manuscript is well-written in clear and concise English. The tables and figures are also appropriate, clear, and well notated. I provide a few suggested edits as noted in a subsequent section.

Response: Many thanks.

- 3. Agree with Reviewer 1 that detailed mathematical derivation can be placed in an appendix. Response: The derivation of the present solution has been moved to the supplementary material, and then the Methodology section is shortened. Please refer to the revised manuscript as attached.
- 4. Your model might be better formulated using alternative parameters (e.g., Depner and Rasmussen,

2017, Hydrodynamics of Time-Periodic Groundwater Flow: Diffusion Waves in Porous Media):

(a) Equation 1 can be written more parsimoniously using:

$$D_r\left[\frac{\partial^2 h}{\partial r^2} + \frac{1}{r}\frac{\partial h}{\partial r} + \alpha \frac{\partial^2 h}{\partial z^2}\right] = \frac{\partial h}{\partial t}$$

where  $D_r = K_r/S_s$  and  $\alpha = K_z/K_r$ , which reduces the number of model parameters from three to two.

(b) Equation 4. The vertical flux at z = b is:

$$q_z = -K_z \frac{\partial h}{\partial z} \tag{4a}$$

which can be defined for DGD conditions using (Boulton, 1954):

$$q_z = \frac{s_y}{\kappa} \int_0^t \frac{\partial h}{\partial \tau} \exp \frac{-(t-\tau)}{\kappa} d\tau$$
(4b)

where  $\kappa = 1/\epsilon$ , which has units of time rather than inverse time. Note that Eqn 4b reduces to IGD conditions as  $\kappa \to \infty$ :

$$q_z = S_y \frac{\partial h}{\partial t} \tag{4c}$$

Solving for the boundary gradient gives:

$$\frac{\partial h}{\partial z} = -\frac{1}{\kappa C_y} \int_0^t \frac{\partial h}{\partial \tau} e^{-\frac{r-\tau}{\kappa}} d\tau$$
(4d)

where  $C_y = K_z/S_y$ , with units of L/T.

- (c) Note that  $D_r$  and  $\alpha$  are domain parameters defined by Eqn 1,  $K_r$  is a boundary parameter defined by Eqn 3, and  $C_y$  and  $\kappa$  are boundary parameters defined by Eqn 4, where boundary parameters describe the aquifer characteristics on or near the boundary, and domain parameters describe the average characteristics within the interior of the aquifer. All other parameters (i.e.,  $K_z$ ,  $S_s$ ,  $S_y$ ) are hybrid domain-boundary parameters that are a composite of both boundary and domain characteristics.
- (d) Dimensionless parameters in Eqn 7 can now be defined using:

$$\bar{t} = t(D_r/r_w^2) \qquad \bar{P} = P(D_r/r_w^2) \qquad \gamma = \omega(r_w^2/D_r) \qquad \mu = \alpha(r_w^2/b^2) \qquad a_1 = b/(\kappa C_y)$$

Response: Thank for the suggestions. The present model is reformulated as suggested. Please refer to

the revised manuscript as attached.

**Suggested Edits**

1. Title, suggest removing "Consider the"

Response: Done as suggested.

2. Lines 22-24, suggest removing "without net water extraction" because a periodic test can be superimposed on a steady test.

Also, "Oscillatory pumping tests (OPT) provide an alternative to constant-head and constant-rate tests for determining aquifer hydraulic parameters, with many analytical models available for parameter determination."

Response: Thanks for the comments. The sentence is rewritten as "Oscillatory pumping tests (OPTs) provide an alternative to constant-head and constant-rate pumping tests for determining aquifer hydraulic parameters when OPT data are analyzed based on an associated analytical model coupled with an optimization approach." (Page 2, lines 23 – 25 of the revised manuscript).

 Lines 30-31, suggest revising to "The solution is derived using the Laplace, finite-integral, and Weber transforms."

Response: Done as suggested.

4. Line 37, suggest explaining "certain time shift" here and subsequently.

Response: The phrase "certain time shift" in the text is changed to "a time shift".

5. Lines 56-58, suggest noting that periodic signals (depending on frequency) are likely to be observable at far greater distances than constant pumping because the signal-to-noise ratio for periodic testing is smaller due to the lack of noise at the testing frequency, unless there is

interference from natural or artificial sources, such as solar and lunar periodicities.

Response: Thanks for the comment. The phrase "the problem of signal attenuation in remote distance from the pumping well" has been removed.

Line 71, suggest explaining "certain period" here and subsequently.
 Response: The phrase "a certain period" is replaced by "a late period".

7. Line 121, first reference to a partially penetrating pumping well; suggest highlighting in the abstract and introduction.

Response: The phrase "the rim of a finite-radius well" in the abstract is changed to "the rim of a partially screened well" (Page 2, line 31) and the phrase "the pumping well" in the Introduction section is replaced by "the partially screened well". (Page 5, line 96).

- 8. Line 165, suggest capitalizing "Section" here and subsequently (it's a proper noun).
- 9. Line 300, suggest capitalizing "Solution" here and subsequently (it's a proper noun).

Response: Done as suggested.

10. Line 326, Figure 2 is the most interesting aspect of this manuscript; suggest explaining how period affects this plot. What happens when *P* is longer or shorter than ε (or κ = 1/ε, with units of time)? I suspect that a *P* ≫ κ will provide an estimate of Sy (i.e., late-time), while *P* ≪ κ gives Ss (early time). Is it possible to have a dimensionless ratio of *P*/κ?

Response: Thanks for the comment. Figure 2 has been redrawn and also shown below. The associated section is rewritten as follows:

**"3.1. Delayed gravity drainage**

Previous analytical models for OPT consider either confined flow (e.g., Rasmussen et al., 2003) or unconfined flow with IGD effect (e.g., Dagan and Rabinovich, 2014). Little attention has been paid to the consideration of the DGD effect. This section addresses the diffrence among these three models. Figure 2 shows the curve of the dimensionless amplitude  $\bar{A}_t$  at  $(\bar{r}, \bar{z}) = (1, 1)$  of Solution 1 versus the dimensionless parameter  $a_1$  related to the DGD effect. The transient head fluctuations are plotted based on Solution 1 with  $a_1 = 10^{-2}$ , 1, 10, 500, Solution 2 for IGD and Solution 3 for confined flow. Define the relative error as

$$RE = |\bar{A}_t' - \bar{A}_t| / \bar{A}_t \tag{28}$$

where  $\bar{A}'_t$  is the dimensionless amplitude predicted by Solution 2 for the case of  $a_1 = 500$  or Solution 3 for the case of  $a_1 = 10^{-2}$ . The curves of the *RE* versus the period of oscillatory pumping rate (i.e., *P*) for these two cases are displayed. The range of  $P \le 10^5$ s (1.16 d) contains most practical applications of OPT. When  $10^{-2} \le a_1 \le 500$ , the  $\bar{A}_t$  gradually decreases with  $a_1$  to the trough and then increases to the ultimate value of  $\bar{A}_t = 1.79 \times 10^{-2}$ . The DGD, in other words, causes an effect. When  $a_1 < 10^{-2}$ , Solutions 1 and 3 agree on the predicted heads; the *RE* is below 1% for  $P < 10^4$ s (2.78 h), indicating the unconfined aquifer with the DGD effect behaves like confined aquifer and the water table can be regarded as a no-flow boundary when  $a_1 < 10^{-2}$  and  $P < 10^4$ s. When  $a_1 > 500$ , the head fluctuations predicted by both Solutions 1 and 2 are identical; the largest *RE* is about 0.45%, indicating the DGD effect is ignorable and Eq. (4b) reduces to (4a) for the IGD condition. This conclusion is applicable for any magnitude of *P* in spite of  $P > 10^5$ s."

(Pages 13 - 14, lines 280 - 300)

**11. Line 445, suggest explaining "certain trough".**

Response: The phrase "certain trough" is replaced by "trough".

12. Table 2, suggest providing estimated domain ( $D_r$ ,  $\alpha$ ), boundary ( $K_r$ ,  $C_y$ ,  $\kappa$ ), and hybrid ( $K_z$ ,  $S_s$ ,  $S_y$ ) parameters along with their individual standard errors. You might also provide the estimates from Rasmussen et al (2003).

Response: Thanks for the suggestion. Table 2 is rewritten and given at the end of this reply. In order to

compare the parameters reported in Rasmussen et al. (2003), we add two sentences, also given below, in the associated text.

"The estimates of *T*, *S* and  $D_r$  given in Rasmussen et al. (2003) are also presented." (Page 17, lines 381 - 382)

"In addition, the difference in *T*, *S* or *Dr* estimated by Solution 6 and those by the Rasmussen et al. (2003) solution may be attributed to the fact that their solution assumes isotropic hydraulic conductivity (i.e.,  $K_r = K_z$ )." (Page 18, lines 393 - 396)

In addition, two sentences in the same paragraph are rewritten as:

"The result shows the estimated Sy is very small, and the estimated T and S by Solution 3, 6 or the Rasmussen et al. (2003) solution for confined flow are close to those by Solution 4 or 5 for unconfined flow, indicating that the unconfined flow induced by the OPT in the Surficial Aquifer is negligibly small." (Page 17, lines 382 - 385)

"On the other hand, transient Solution 3 gives smaller SEEs than PSS Solution 6 or the Rasmussen et al. (2003) solution for the Barnwell-McBean Aquifer and better fits to the observed data at the early pumping periods as shown in Fig. 7." (Page 18, lines 396 - 398)

**References**

- Dagan, G. and Rabinovich, A.: Oscillatory pumping wells in phreatic, compressible, and homogeneous aquifers, Water Resour. Res., 50(8), 7058–7066, 2014.
- Rasmussen, T. C., Haborak, K. G., and Young, M. H.: Estimating aquifer hydraulic properties using sinusoidal pumping at the Savannah River site, South Carolina, USA, Hydrogeol. J., 11(4), 466– 482, 2003.

Figure 2. Influence of delayed gravity drainage on the dimensionless amplitude  $\bar{A}_t$  and transient head  $\bar{h}$  at  $\bar{r} = 1$ ,  $\bar{z} = 1$  predicted by Solution 1 for different magnitudes of  $a_1$  related to the influence.

| Observation
well     | Solution                       | $T (m^2/s)$           | S                     | $D_r (\mathrm{m}^2/\mathrm{s})$ | $K_z$ (m/s)           | $S_y$                 | $C_y$ (m/s)           | α    | κ (s)  | SEE   | ME                     |
|-------------------------|--------------------------------|-----------------------|-----------------------|---------------------------------|-----------------------|-----------------------|-----------------------|------|--------|-------|------------------------|
| Surficial Aquifer       |                                |                       |                       |                                 |                       |                       |                       |      |        |       |                        |
| 101D                    | Solution 3 a | $9.27 \times 10^{-4}$ | $2.44\times10^{-3}$   | 0.380                           | -                     | -                     | -                     | -    | -      | 0.018 | $-5.56 \times 10^{-3}$ |
|                         | Solution 6 b        | $9.18 \times 10^{-4}$ | $2.33 \times 10^{-3}$ | 0.393                           | -                     | -                     | -                     | -    | -      | 0.018 | $-2.20 \times 10^{-4}$ |
|                         | Solution 4 c        | $4.61\times10^{-4}$   | $3.95 \times 10^{-3}$ | 0.117                           | $7.38 \times 10^{-6}$ | $2.23 \times 10^{-3}$ | $3.31 \times 10^{-3}$ | 0.10 | 94.34  | 0.018 | $-2.20\times10^{-4}$   |
|                         | Solution 5 c        | $5.25 \times 10^{-4}$ | $1.09 \times 10^{-3}$ | 0.482                           | $2.61 \times 10^{-5}$ | $5.49 \times 10^{-3}$ | $4.75\times10^{-3}$   | 0.31 | -      | 0.019 | $-2.30\times10^{-4}$   |
|                         | Rasmussen et al. $(2003)^b$    | $2.17 \times 10^{-3}$ | $1.35 \times 10^{-4}$ | 16.074                          | -                     | -                     | -                     | -    | -      | 0.018 | $-2.20 \times 10^{-4}$ |
| 102D                    | Solution 3 a | $9.13 \times 10^{-4}$ | $1.76 \times 10^{-3}$ | 0.519                           | -                     | -                     | -                     | -    | -      | 0.010 | $-4.38 \times 10^{-3}$ |
|                         | Solution 6 b        | $9.17 	imes 10^{-4}$  | $1.67 \times 10^{-3}$ | 0.549                           | -                     | -                     | -                     | -    | -      | 0.011 | $9.57 \times 10^{-4}$  |
|                         | Solution 4 c        | $9.57 \times 10^{-5}$ | $7.85 \times 10^{-4}$ | 0.122                           | $3.68 \times 10^{-6}$ | $4.95 \times 10^{-3}$ | $7.43 \times 10^{-4}$ | 0.24 | 420.17 | 0.011 | $9.57 \times 10^{-4}$  |
|                         | Solution 5 c        | $9.49 \times 10^{-5}$ | $3.25 \times 10^{-4}$ | 0.292                           | $4.67 \times 10^{-6}$ | $4.68\times10^{-3}$   | $9.98 \times 10^{-4}$ | 0.31 | -      | 0.011 | $9.50 \times 10^{-4}$  |
|                         | Rasmussen et al. $(2003)^b$    | $2.27 \times 10^{-3}$ | $2.28 \times 10^{-4}$ | 9.956                           | -                     | -                     | -                     | -    | -      | 0.011 | $9.57 \times 10^{-4}$  |
| Barnwell-McBean Aquifer |                                |                       |                       |                                 |                       |                       |                       |      |        |       |                        |
| 201C                    | Solution 3 a | $5.86 \times 10^{-5}$ | $7.07 \times 10^{-4}$ | 0.083                           | -                     | -                     | -                     | -    | -      | 0.232 | 0.046                  |
|                         | Solution 6 b        | $6.03 \times 10^{-5}$ | $6.54 \times 10^{-4}$ | 0.092                           | -                     | -                     | -                     | -    | -      | 0.363 | 0.281                  |
|                         | Rasmussen et al. $(2003)^b$    | $6.90 \times 10^{-5}$ | $4.74 	imes 10^{-4}$  | 0.150                           | -                     | -                     | -                     | -    | -      | 0.363 | 0.281                  |

Table 2. Hydraulic parameters estimated by the present solution and the Rasmussen et al. (2003) solution for OPT data from the Savannah River site

*a* transient confined flow

b PSS confined flow

c PSS unconfined flow

**Reply to Reviewer 3**

General comments:

Authors describe new analytical solutions to oscillatory pumping tests, applied to data collected at the Savanna River Site in South Carolina, USA, and published by Rasmussen et al. (2003). The solutions extend those published earlier by several authors, by now including delayed gravity drainage, finite radius pumping wells and initial conditions in the well bore. The solutions were well described and the writing was clear.

In general, I agree with the others reviewers that much of the in-depth derivations of the solutions can be moved to the appendix or supplemental section so that the authors could spend more time on the geology and results of the study. As presented, only about 1.5 pages of the manuscript was devoted the testing of the solutions with real field data. Moving derivations to the appendix would also improve readability of the manuscript, which as presented is extremely dense and likely would appeal to a very few number of applied mathematicians and/or hydrologists. Simplify the presentation of the material, and more readers will take the time to read the manuscript, and cite the work.

Response: Thanks for the suggestion. The derivation of the present solution has been moved to the supplementary material, and then the Methodology section is shortened. Please refer to the revised manuscript as attached.

Specific comments:

L340 – Yeh and Chang (2013) not included in the references Response: The reference of Yeh and Chang (2013) is added.

L377 – replace 'to' with 'with'

Response: Done as suggested.

L385 - check sentence that begins on this line; it is unclear as written and needs some clarification

Response: The sentence is rewritten and given below:

"The phase of Solution 6 (i.e.,  $\phi_s = 1.50$  rad for panel (a) and 1.33 rad for (b)) can be replaced by the phase of Solution 3 (i.e.,  $\phi_t = 1.64$  rad for (a) and 1.81 rad for (b)) so that the  $\bar{h}_{SHM}$  prediction of Solutions 3 is identical to the  $\bar{h}_s$  prediction of Solution 6." (Page 16, lines 351 - 354 of the revised manuscript)

L425 - replace 'researches' with 'research outcomes'

L443 – re-write portion of sentence as 'the effect of DGD on head fluctuations should be considered.' Response: Thanks, they are modified as suggested.

[revised manuscript text omitted]

 (17)

- 207 where  $\overline{H}(\overline{r},\overline{z})$  is a space function of  $\overline{r}$  and  $\overline{z}$ . Define a PSS IGD model as Eqs. (8) (13)
- 208 excluding (9), (11b) and replacing  $\sin(\gamma \bar{t})$  in (10) by  $e^{i\gamma \bar{t}}$ . Substituting Eq. (17) and
- 209  $\partial \bar{h}_{s} / \partial \bar{t} = \text{Im}(i\gamma \bar{H}(\bar{r}, \bar{z}) e^{i\gamma \bar{t}})$  into the model results in

$$\qquad \frac{\partial^2 \overline{H}}{\partial \overline{r}^2} + \frac{1}{\overline{r}} \frac{\partial \overline{H}}{\partial \overline{r}} + \mu \frac{\partial^2 \overline{H}}{\partial \overline{z}^2} = i\gamma \overline{H}$$
(18)

211
$$\frac{\partial \bar{H}}{\partial \bar{r}} = \begin{cases} 1 \text{ for } \bar{z}_l \le \bar{z} \le \bar{z}_u \\ 0 \text{ outside screen interval} \end{cases} \text{ at } \bar{r} = 1$$
(19)

212
$$\frac{\partial \overline{H}}{\partial \overline{z}} = -ia\gamma \overline{H}$$
 at  $\overline{z} = 1$  for IGD (20)

213
$$\frac{\partial \overline{H}}{\partial \overline{z}} = 0$$
 at  $\overline{z} = 0$  (21)

$$\quad \lim_{\bar{r} \to \infty} \bar{H} = 0 \tag{22}$$

The resultant model is independent of  $\bar{t}$ , indicating the analytical solution of  $\bar{H}(\bar{r}, \bar{z})$  is tractable. Similarly, consider a PSS DGD model that equals the PSS IGD model but replaces (11a) by (11b). Substituting Eq. (17) into the result yields a model that depends on  $\bar{t}$ , indicating the solution  $\bar{h}_s$  to the PSS DGD model is not tractable.

219 The Weber transform, defined in Eq. (B.1) of the supporting material, may be considered

as a Hankel transform with a more general kernel function. It can be applied to diffusion-type problems with a radial-symmetric region from a finite distance to infinity. For groundwater flow problems, it can be used to develop the analytical solution for the flow equation with a Neumann boundary condition specified at the rim of a finite-radius well (e.g., Lin and Yeh, 2017; Povstenko, 2015). Taking the transform and the formula of  $e^{i\gamma\bar{t}} = \cos(\gamma\bar{t}) + i\sin(\gamma\bar{t})$

225 to solve Eqs. (18) - (22) yields the solution of  $\bar{h}_s$  expressed as

226
$$\bar{h}_{s}(\bar{r},\bar{z},\bar{t}) = \bar{A}_{s}(\bar{r},\bar{z})\cos(\gamma t - \phi_{s}(\bar{r},\bar{z}))$$
(23a)

227
$$\bar{A}_{s}(\bar{r},\bar{z}) = \sqrt{a_{s}(\bar{r},\bar{z})^{2} + b_{s}(\bar{r},\bar{z})^{2}}$$
 (23b)

228
$$a_s(\bar{r},\bar{z}) = \operatorname{Re}(\bar{H}(\bar{r},\bar{z}))$$
 (23c)

229
$$b_{\rm s}(\bar{r},\bar{z}) = \operatorname{Im}(\bar{H}(\bar{r},\bar{z}))$$
 (23d)

230
$$\phi_s(\bar{r},\bar{z}) = \cos^{-1}(b_s(\bar{r},\bar{z})/A_s(\bar{r},\bar{z}))$$
 (23e)

231
$$\overline{H}(\bar{r}, \bar{z}) = \begin{cases} \int_0^\infty \widetilde{H}_u \,\xi \,\Omega \,d\xi & \text{for } \bar{z}_u < \bar{z} \le 1 \\ \int_0^\infty \widetilde{H}_m \,\xi \,\Omega \,d\xi & \text{for } \bar{z}_l \le \bar{z} \le \bar{z}_u \\ \int_0^\infty \widetilde{H}_l \,\xi \,\Omega \,d\xi & \text{for } 0 \le \bar{z} < \bar{z}_l \end{cases}$$
(23f)

232
$$\Omega = \left( J_0(\xi \bar{r}) Y_1(\xi) - Y_0(\xi \bar{r}) J_1(\xi) \right) / \left( J_1^2(\xi) + Y_1^2(\xi) \right)$$
(23g)

with the Bessel functions of the first kind of order zero  $J_0(-)$  and one  $J_1(-)$  as well as the second kind of order zero  $Y_0(-)$  and one  $Y_1(-)$ ,

235
$$\begin{cases} \widetilde{H}_{u} = \widetilde{H}_{p}(c_{1} \exp(\lambda_{w} \bar{z}) + c_{2} \exp(-\lambda_{w} \bar{z})) \text{ for } \bar{z}_{u} < \bar{z} \leq 1\\ \widetilde{H}_{m} = \widetilde{H}_{p}(c_{3} \exp(\lambda_{w} \bar{z}) + c_{4} \exp(-\lambda_{w} \bar{z}) - 1) \text{ for } \bar{z}_{l} \leq \bar{z} \leq \bar{z}_{u}\\ \widetilde{H}_{l} = \widetilde{H}_{p}c_{5}(\exp(\lambda_{w} \bar{z}) + \exp(-\lambda_{w} \bar{z})) \text{ for } 0 \leq \bar{z} < \bar{z}_{l} \end{cases}$$
(23h)

236
$$c_1 = -e^{-\lambda_w}(\lambda_w - \sigma)(\sinh(\bar{z}_l\lambda_w) - \sinh(\bar{z}_u\lambda_w))/D$$
 (23i)

237
$$c_2 = -e^{\lambda_w} (\lambda_w + \sigma) (\sinh(\bar{z}_l \lambda_w) - \sinh(\bar{z}_u \lambda_w)) / D$$
(23j)

238
$$c_{3} = \frac{e^{-(1+\bar{z}_{l}+\bar{z}_{u})\lambda_{w}}}{2D} \left( \sigma \left( e^{(2+\bar{z}_{l})\lambda_{w}} + e^{\bar{z}_{u}\lambda_{w}} - e^{(2\bar{z}_{l}+\bar{z}_{u})\lambda_{w}} \right) + (\sigma - \lambda_{w}) e^{(\bar{z}_{l}+2\bar{z}_{u})\lambda_{w}} + 239 \lambda_{w} \left( e^{(2+\bar{z}_{l})\lambda_{w}} - e^{\bar{z}_{u}\lambda_{w}} + e^{(2\bar{z}_{l}+\bar{z}_{u})\lambda_{w}} \right) \right)$$
(23k)

240
$$c_4 = \frac{e^{-(1+\bar{z}_l+\bar{z}_u)\lambda_w}}{2D} \Big( (\sigma - \lambda_w) e^{(\bar{z}_l+2\bar{z}_u)\lambda_w} + (\sigma + \lambda_w) (e^{(2+\bar{z}_l)\lambda_w} - e^{(2+\bar{z}_u)\lambda_w} + (\sigma + \lambda_w)) \Big) \Big) \Big) \Big| c_1 = \frac{1}{2D} \Big( (\sigma - \lambda_w) e^{(\bar{z}_l+2\bar{z}_u)\lambda_w} + (\sigma + \lambda_w) (e^{(2+\bar{z}_l)\lambda_w} - e^{(2+\bar{z}_u)\lambda_w} + (\sigma + \lambda_w)) \Big) \Big) \Big) \Big| c_1 = \frac{1}{2D} \Big( (\sigma - \lambda_w) e^{(\bar{z}_l+2\bar{z}_u)\lambda_w} + (\sigma + \lambda_w) (e^{(2+\bar{z}_l)\lambda_w} - e^{(2+\bar{z}_u)\lambda_w} + (\sigma + \lambda_w)) \Big) \Big) \Big| c_1 = \frac{1}{2D} \Big( (\sigma - \lambda_w) e^{(\bar{z}_l+2\bar{z}_u)\lambda_w} + (\sigma + \lambda_w) (e^{(2+\bar{z}_l)\lambda_w} - e^{(2+\bar{z}_u)\lambda_w} + (\sigma + \lambda_w)) \Big) \Big) \Big| c_1 = \frac{1}{2D} \Big| c_1 = \frac{1}{2D} \Big| c_1 = \frac{1}{2D} \Big| c_2 = \frac{1}{2D} \Big| c_1 = \frac{1}{2D} \Big| c_2 = \frac{1}{2D} \Big$$

$$\quad e^{(2+2\bar{z}_l+\bar{z}_u)\lambda_w})\Big) \tag{231}$$

242
$$c_{5} = \frac{1}{2D} e^{-(1+\bar{z}_{l}+\bar{z}_{u})\lambda_{w}} (e^{\bar{z}_{l}\lambda_{w}} - e^{\bar{z}_{u}\lambda_{w}}) ((\lambda_{w} - \sigma)e^{(\bar{z}_{l}+\bar{z}_{u})\lambda_{w}} + (\lambda_{w} + \sigma)e^{2\lambda_{w}})$$
(23m)

where  $\lambda_w^2 = (\xi^2 + i\gamma)/\mu$ ,  $\sigma = i\gamma a$ ,  $\tilde{H}_p = 2/(\pi\mu\xi\lambda_w^2)$  and  $D = 2(\sigma\cosh\lambda_w + \lambda_w\sinh\lambda_w)$ , and Re(-) is the real part of a complex number. Again, one can refer to the supporting material for the derivation of the solution. Eq. (23a) indicates SHM for the response of the hydraulic head at any point to oscillatory pumping. Note that Eq. (23f) reduces to  $\bar{H}(\bar{r},\bar{z}) = \int_0^\infty \tilde{H}_m \xi \Omega d\xi$  for a fully screened well when  $\bar{z}_l = 0$  and  $\bar{z}_u = 1$ .

**248 **2.6.** Pseudo-steady state solution for confined aquifers**

Applying the finite Fourier cosine transform to the model, Eqs. (18) – (22) with  $S_y = 0$  (i.e., a = 0) for the confined condition, leads to an ordinary differential equation with two boundary conditions. With solving the boundary-value problem, the solution of  $\bar{h}_s$  for confined aquifers can be expressed as Eqs. (23a) - (23e) with  $\bar{H}(\bar{r}, \bar{z})$  defined as

253
$$\overline{H}(\bar{r},\bar{z}) = -2\sum_{m=0}^{\infty} \frac{K_0(\bar{r}\lambda_m)}{\lambda_m K_1(\lambda_m)} \times \begin{cases} 0.5(\bar{z}_u - \bar{z}_l) \text{ for } m = 0\\ \frac{\cos(m\pi\bar{z})}{m\pi}(\sin(\bar{z}_u m\pi) - \sin(\bar{z}_l m\pi)) \text{ for } m > 0 \end{cases}$$
 (24)

where  $\lambda_m^2 = \gamma i + \mu (m\pi)^2$ . The derivation of Eq. (24) is also listed in the supporting material. For a fully screened well (i.e.,  $\bar{z}_u = 1$ ,  $\bar{z}_l = 0$ ), the first term of the series (i.e., m = 0) remains and the others equal zero because of  $\sin(\bar{z}_u m\pi) - \sin(\bar{z}_l m\pi) = 0$ . The result is independent of dimensionless elevation  $\bar{z}$ , indicating the confined flow is only horizontal.

**258 **2.7. Special cases of the present solution**

Table 1 classifies the present solution (i.e., Solution 1) and its special cases (i.e., Solutions 2 to 6) according to transient or PSS flow, unconfined or confined aquifer, and IGD or DGD. Each of Solutions 1 to 6 reduces to a special case for fully screened well. Existing analytical solutions can be regarded as special cases of the present solution as discussed in Section 3.4 (e.g., Black and Kipp, 1981; Rasmussen et al., 2003; Dagan and Rabinovich, 2014).

**264 **2.8. Sensitivity analysis**

Sensitivity analysis evaluates hydraulic head variation in response to the change in each of  $K_r$ ,  $K_z$ ,  $S_s$ ,  $S_y$ ,  $\omega$ , and  $\varepsilon$ . The normalized sensitivity coefficient can be defined as (Liou and Yeh, 1997)

$$268 S_i = P_i \frac{\partial X}{\partial P_i} (25)$$

where  $S_i$  is the sensitivity coefficient of *i*th parameter;  $P_i$  is the magnitude of the *i*th input parameter; *X* represents the present solution in dimensional form. Eq. (25) can be approximated as

272
$$S_i = P_i \frac{X(P_i + \Delta P_i) - X(P_i)}{\Delta P_i}$$
(26)

273 where  $\Delta P_i$ , a small increment, is chosen as  $10^{-3}P_i$ .

**274 **3. Results and Discussion**

The following sections demonstrate the response of the hydraulic head to oscillatory pumping using the present solution. The default values in calculation are r = 0.05 m, z = 5 m, b = 10 m,  $Q = 10^{-3}$  m3/s,  $r_w = 0.05$  m,  $z_u = 5.5$  m,  $z_l = 4.5$  m,  $K_r = 10^{-4}$  m/s,  $K_z = 10^{-5}$  m/s,  $S_s = 10^{-5}$  m-1,  $S_y$  $= 10^{-4}$ ,  $\omega = 2\pi/30$  s-1, and  $\kappa = 100$  s. The corresponding dimensionless parameters and variables are  $\bar{r} = 1$ ,  $\bar{z} = 0.5$ ,  $\bar{z}_u = 0.55$ ,  $\bar{z}_l = 0.45$ ,  $\gamma = 5.24 \times 10^{-5}$ ,  $\mu = 2.5 \times 10^{-6}$ ,  $a = 4 \times 10^5$ ,  $a_1 = 1$  and  $a_2 = 2.5 \times 10^{-6}$ .

**281 **3.1. Delayed gravity drainage**

Previous analytical models for OPT consider either confined flow (e.g., Rasmussen et al., 2003) or unconfined flow with IGD effect (e.g., Dagan and Rabinovich, 2014). Little attention has been paid to the consideration of the DGD effect. This section addresses the diffrence among these three models. Figure 2 shows the curve of the dimensionless amplitude  $\bar{A}_t$  at ( $\bar{r}$ ,  $\bar{z}$ ) = (1, 1) of Solution 1 versus the dimensionless parameter  $a_1$  related to the DGD effect. The transient head fluctuations are plotted based on Solution 1 with  $a_1 = 10^{-2}$ , 1, 10, 500, Solution 2 for IGD and Solution 3 for confined flow. Define the relative error as

289
$$RE = |\bar{A}'_t - \bar{A}_t| / \bar{A}_t$$
 (27)

where  $\bar{A}'_t$  is the dimensionless amplitude predicted by Solution 2 for the case of  $a_1 = 500$ 290 or Solution 3 for the case of  $a_1 = 10^{-2}$ . The curves of the *RE* versus the period of oscillatory 291 pumping rate (i.e., P) for these two cases are displayed. Th